# Modulation of alpha oscillations by attention is predicted by hemispheric asymmetry of subcortical regions

Tara Ghafari[1]*, Cecilia Mazzetti[1], Kelly Garner[2], Tjerk Gutteling[1,3], Ole Jensen[1]*

[1]Centre for Human Brain Health, School of Psychology, University of Birmingham, Birmingham, United Kingdom; [2]School of Psychology, University of New South Wales, Kensington, Australia; [3]CERMEP-Imagerie du Vivant, MEG Department, Lyon, France

*For correspondence:
t.ghafari@bham.ac.uk (TG);
o.jensen@bham.ac.uk (OJ)

**Abstract** Evidence suggests that subcortical structures play a role in high-level cognitive functions such as the allocation of spatial attention. While there is abundant evidence in humans for posterior alpha band oscillations being modulated by spatial attention, little is known about how subcortical regions contribute to these oscillatory modulations, particularly under varying conditions of cognitive challenge. In this study, we combined MEG and structural MRI data to investigate the role of subcortical structures in controlling the allocation of attentional resources by employing a cued spatial attention paradigm with varying levels of perceptual load. We asked whether hemispheric lateralization of volumetric measures of the thalamus and basal ganglia predicted the hemispheric modulation of alpha-band power. Lateral asymmetry of the globus pallidus, caudate nucleus, and thalamus predicted attention-related modulations of posterior alpha oscillations. When the perceptual load was applied to the target and the distractor was salient caudate nucleus asymmetry predicted alpha-band modulations. Globus pallidus was predictive of alpha-band modulations when either the target had a high load, or the distractor was salient, but not both. Finally, the asymmetry of the thalamus predicted alpha band modulation when neither component of the task was perceptually demanding. In addition to delivering new insight into the subcortical circuitry controlling alpha oscillations with spatial attention, our finding might also have clinical applications. We provide a framework that could be followed for detecting how structural changes in subcortical regions that are associated with neurological disorders can be reflected in the modulation of oscillatory brain activity.

## eLife assessment

This study by Ghafari et al. tackles a question relevant for the field of attention as it connects structural differences in subcortical regions with oscillatory modulations during attention allocation. Using a combination of Magnetoencephalography (MEG) and magnetic resonance imaging (MRI) data in human subjects, the **valuable** results show that inter-individual differences in the lateralisation of alpha oscillations are explained by asymmetry of subcortical brain regions. The strength of evidence is deemed **convincing** in line with current state-of-the-art.

## Introduction

The visual world provides more sensory information than we can be aware of at any given moment. Thus, our brains must prioritise goal-relevant over distracting information. A rich body of research shows that the brain amplifies goal-relevant inputs, and suppresses non-relevant inputs by a process referred to as selective attention (***Nobre and Kastner, 2014***; ***Desimone and Duncan, 1995***; ***Moran***

*and Desimone, 1985*). There is ample evidence for top-down control of neocortical regions associated with sensory processing when information is prioritized (*Nobre and Kastner, 2014*; *Corbetta and Shulman, 2002*; *Noudoost et al., 2010*). The dorsal attention network, which consists of the intraparietal sulcus/superior parietal lobule, and the frontal eye fields, is the most predominant network associated with the allocation of attention (*Kastner et al., 1999*; *Hopfinger et al., 2000*; *Corbetta et al., 2000*). However, although the role of neocortex for spatial attention and cognitive control has been extensively studied, the contributions of subcortical regions are less well understood. One reason, amongst many others, is that MEG and EEG are not well suited for detecting subcortical activity. Therefore, the present study aims to provide insights into the contribution of the thalamus and basal ganglia in driving top-down spatial attention.

There has been intense focus on the cortical contributions to the top-down control processes, yet there are multiple sources of evidence to suggest that subcortical structures also play an important role in cognitive control. For instance, it has been shown that the pulvinar plays an important role in the modulation of neocortical alpha oscillations associated with the allocation of attention (*Kastner et al., 2020*). Studies in rats and non-human primates have shown that both the thalamus and superior colliculus, are involved in the control of spatial attention by contributing to the regulation of neocortical activity (*Kastner et al., 2020*; *Krauzlis et al., 2013*; *Krauzlis et al., 2018*). Notably, when the largest nucleus of the thalamus, the pulvinar, was inactivated after muscimol infusion, the monkey's ability to detect color changes in attended stimuli was lowered. This behavioral deficit occurred when the target was in the receptive field of V4 neurons that were connected to lesioned pulvinar (*Zhou et al., 2016*). The basal ganglia play a role in different aspects of cognitive control, encompassing attention (*van Schouwenburg et al., 2010*; *Nakajima et al., 2019*), behavioural output (*Moolchand et al., 2022*), and conscious perception (*Slagter et al., 2017*). Moreover, the basal ganglia contribute to visuospatial attention by linking with cortical regions like the prefrontal cortex via the thalamus. Anatomical tracing studies on selective attention and distractor suppression point to a key role of prefrontal-basal ganglia-thalamus pathway whereby sensory thalamic activity is regulated by prefrontal cortex via basal ganglia (*Nakajima et al., 2019*). Furthermore, fMRI studies in humans demonstrated increased activation in basal ganglia when covert attention was reallocated. Additionally, dynamic causal modelling has shown that the basal ganglia can modulate the top-down influence of the prefrontal cortex on the visual cortex in a task-dependent manner (*van Schouwenburg et al., 2015*).

In terms of neuronal dynamics, power modulation of oscillatory activity in the alpha band (8–13 Hz) has been proposed to reflect resource allocation between goal-relevant and irrelevant stimuli. This has consistently been shown between studies in EEG and MEG in which attention is allocated to the left or right hemifield. Such studies typically find an alpha power decrease in the hemisphere contralateral to the attended stimuli complemented by a relative increase in alpha power in the other hemisphere associated with unattended stimuli (*Okazaki et al., 2014*; *Thut et al., 2006*; *Worden et al., 2000*). It is debated whether the alpha power associated with the unattended stimuli is under task-driven top-down control or rather explained by an indirect control mechanism driven by the engagement of the target (*Jensen, 2023*). The latter notion is aligned with perception load theory that is defined as the perceptual demand of the task or relevant stimulus, according to which the (finite) resources are allocated (*Lavie, 1995*). Indeed, a recent study demonstrated when the target stimulus has a higher perceptual load (e.g. more difficult to perceive), alpha band power increases in ipsilateral regions thus indirectly reflecting distractor suppression (*Gutteling et al., 2022*).

Based on these findings, both oscillatory activity in the alpha band and the activity of subcortical structures are involved in the allocation of attentional resources. The direct relationship between activity in subcortical regions and neocortical oscillations is poorly understood in humans, in part owing to the difficulty in detecting the activity of deep structures using MEG/EEG. One way around this is to instead investigate, the relationship between the volumetric measures of subcortical structures and oscillatory brain activity by combining MRI and electrophysiological measures such as MEG. Using this approach, it was shown that the hemispheric lateralized modulation of alpha oscillations is correlated with the volumetric hemispheric asymmetry of both the globus pallidus and the thalamus (*Mazzetti et al., 2019*). The relationship between the globus pallidus and the modulation of alpha oscillations was demonstrated in the trials where the visual stimuli were associated with high-value (positive or negative) reward valence.

In this study, we aimed to identify a link between the volumetric asymmetries of subcortical structures and the modulation of alpha oscillations in the context of spatial attention without explicit reward-associations. Given the assumed contribution of the basal ganglia to reward-based learning (*Kasanova et al., 2017*; *Cools et al., 2009*; *Hikosaka et al., 2014*; *Fallon and Cools, 2014*), it is perhaps unsurprising to find contributions of the globus pallidus in the paradigms targeting reward valence. What remains to be determined is whether these structures play a more general role in the formation of spatial attention biases. We analyzed MEG and structural data from a previous study (*Gutteling et al., 2022*), in which spatial cues guided participants to covertly attend to one stimulus (target) and ignore the other (distractor). To investigate the relationship between the allocation of attentional resources and mechanisms of neural excitability and suppression, the target load and the visual saliency of the distractor were manipulated using a noise mask. This load/salience manipulation resulted in four conditions that affect the attentional demands of target and distractor. We utilized the hemispheric laterality of subcortical structures and alpha modulation to overcome issues related to individual variations in oscillatory power and head-size. This approach allowed us to relate the hemispheric volumetric asymmetries in thalamus, caudate nucleus, and globus pallidus to the modulation of alpha oscillations when spatial attention is allocated under varying conditions of cognitive challenge. By examining their role in a task without explicit reward, we aim to elucidate the generalizability of the contributions of subcortical structures to spatial attention modulation. Such a finding would implicate a role for the basal ganglia in cognition beyond the well-studied realm of the

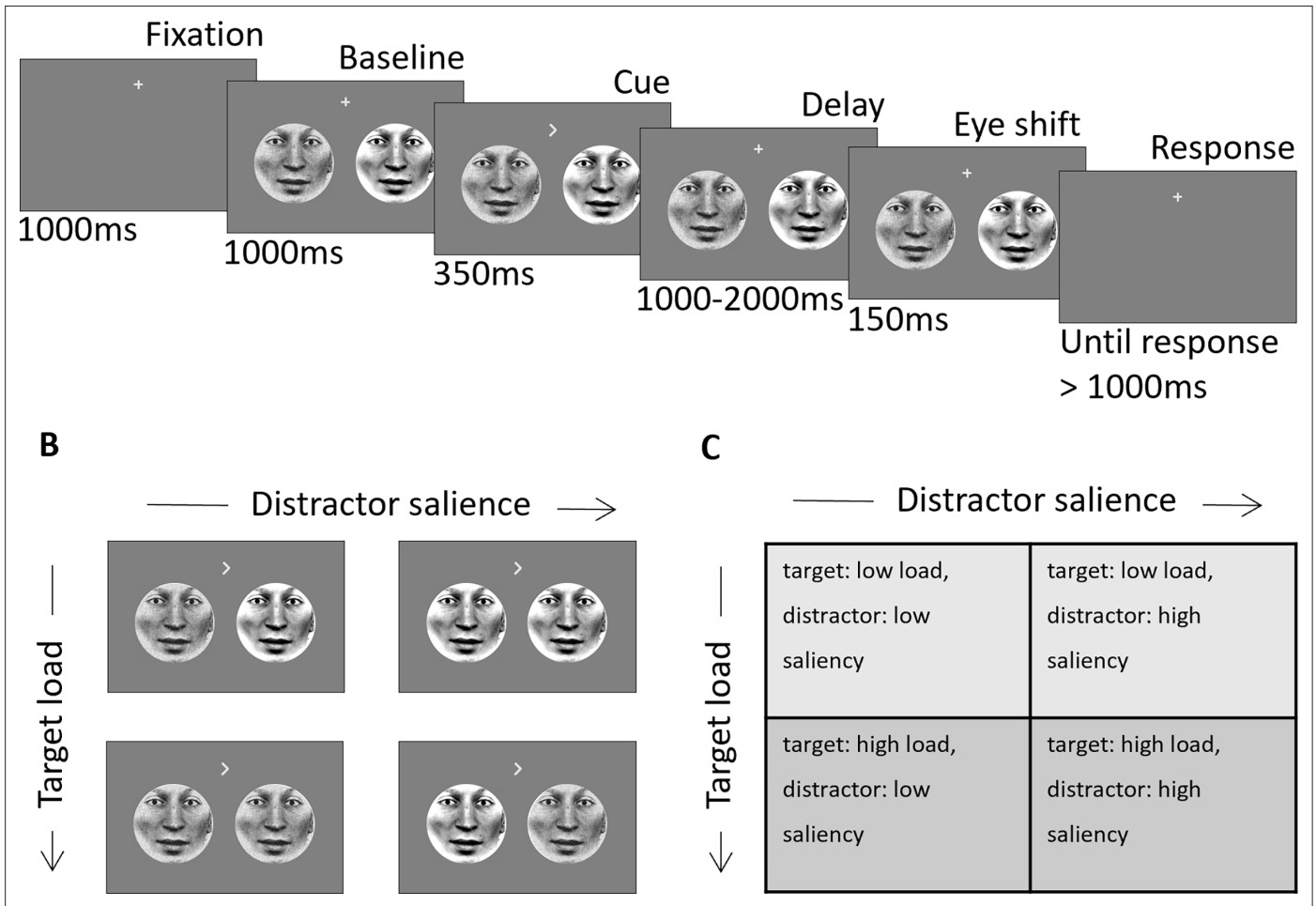

**Figure 1.** Schematic of experimental design. (**A**) Two face stimuli were presented simultaneously in the left and right hemifield. After baseline, a directional cue indicated the location of the target. After a variable delay interval (1000–2000ms) the eye-gaze of each stimulus (independent of the other) shifted randomly to the right or left. Subjects had to indicate the direction of the target eye movement after the delay interval. (**B**) Examples of visual stimuli for each of the four conditions. (**C**) Table with the labels of the four load/salience conditions. Adapted from Figure 1 of *Gutteling et al., 2022*.

estimation of choice values (*Montague et al., 2004*). Specifically, in a prior study (*Mazzetti et al., 2019*), we observed that the contributions of the basal ganglia were most pronounced when the items in question were associated with a reward. Our current findings broaden our understanding of how subcortical structures are involved in modulating alpha oscillations during top-down spatial attention, in the absence of any reward or value associations.

## Results

We investigated the relationship between the volumetric lateralization of subcortical structures estimated from structural MRIs and the hemispheric modulation of alpha oscillations measured by MEG in a spatially cued change detection task. We asked the participants to covertly attend to face-stimuli in the left or right visual field and indicate the direction of a subtle gaze-shift of the attended face (*Figure 1A*). The influences of perceptual load and distractor salience were examined by combining noisy and clear target and distractor stimuli in a 2x2 design (*Figure 1B*).

### Modulation of alpha power with respect to left and right cues

To quantify the anticipatory change in alpha power, we analysed the modulation of power in the –850–0ms interval prior to the target. As expected from a previous report (*Gutteling et al., 2022*), we observed a power decrease contralateral to the cued hemifield and a relative increase ipsilaterally (i.e. an increase contralateral to the distractor, *Figure 2A*) As expected, the magnitude of the modulation index (MI($\alpha$)) reflecting the relative difference in alpha power when attending left versus right, gradually decreased and increased over respectively the left and right hemisphere until target onset (*Figure 2B*). We then identified symmetric clusters of sensors (5 over each hemisphere) that showed the highest modulation of alpha power (*Figure 2C*) and focused the subsequent analyses on these sensors of interest.

### Hemispheric asymmetry of subcortical regions

Next, we computed the hemispheric lateralization modulation of alpha power (HLM($\alpha$)) in each individual. We did so using the HLM($\alpha$) index which quantifies how strongly the alpha power in the left hemisphere is modulated by attention with respect to alpha power modulations in the right hemisphere.

The histogram in *Figure 3A* illustrates the distribution of HLM($\alpha$) in all participants. HLM($\alpha$) indices range from ~–0.15 to 0.15 and are normally distributed around zero before target onset (Shapiro-Wilk, W=0.966, p-value = 0.3895).

We then calculated the hemispheric lateralized volumes of the seven subcortical structures, as illustrated in *Figure 3B* (thalamus, caudate nucleus, putamen, globus pallidus, hippocampus, amygdala, and nucleus accumbens) using the FIRST algorithm on the MRI data. Thalamus (mean ± std = –0.0123±0.0121, p-value <0.000), putamen (mean ± std = –0.0149±0.0285, p-value = 0.004) and nucleus accumbens (mean ± std = –0.1141±0.0746, p-value <0.000) have significantly negative LV values (i.e. left lateralization) whereas the caudate nucleus is right lateralized (mean ± std = 0.0115±0.0285, p-value = 0.021; *Figure 3B*). Globus pallidus, hippocampus, and amygdala did not show any robust volume lateralization.

### Relationship between subcortical regions and hemispheric alpha lateralization

To test whether the individual hemispheric asymmetries in subcortical grey matter relate to variability in HLM(*a*), we subjected the MEG and MRI data to a General Linear Model (GLM). In this model, the individual HLM($\alpha$) values was the dependent variable, and the individual hemispheric lateralization volumes (LV) of the subcortical region were the explanatory variables. To discover the best set of subcortical structures that predict HLM($\alpha$) we used all possible combinations of regressors (LV) and selected the winning model based on lowest Akaike Information Criterion (AIC) scores. The winning model constituted of thalamus, caudate nucleus and globus pallidus and is defined as:

$$HLM\left(\alpha\right) \sim \beta_0 + \beta_1 LV_{Th} + \beta_2 LV_{CN} + \beta_3 LV_{GP} + \varepsilon$$

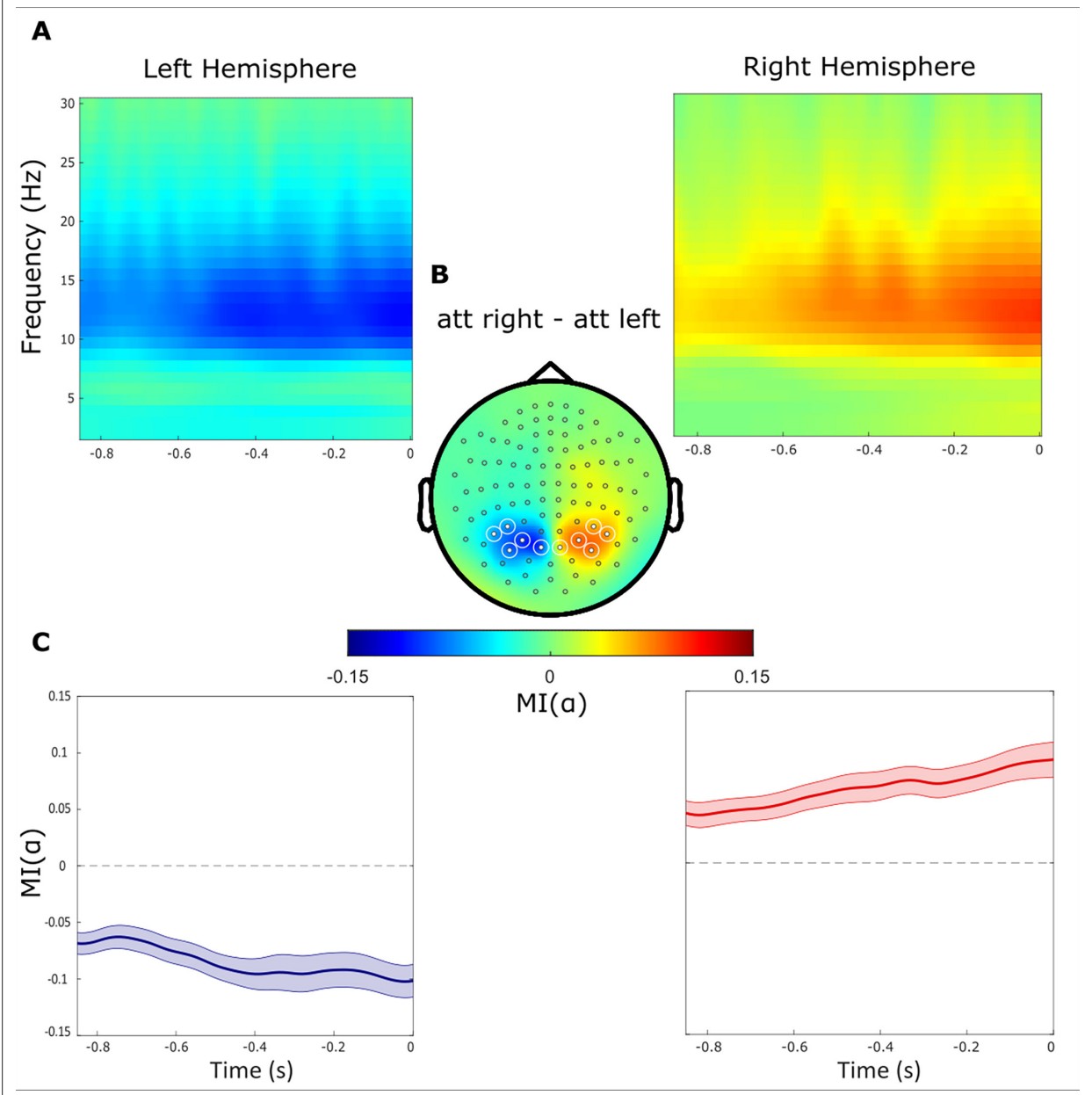

**Figure 2.** Alpha power decreases contralaterally and increases ipsilaterally with respect to the cued hemifield. (**A**) Time-frequency representations of power demonstrate the difference between attended right versus left trials (t=0 indicate the target onset). (**B**) Topographical plot of the relative difference between attend right versus left trials. Regions of Interest sensors (ROIs) are marked with white circles. (**C**) The alpha band modulation MI(α) averaged over ROI sensors within the left and right hemispheres, respectively. The absolute MI(α) increased gradually during the delay interval until the onset of the target stimuli.

where HLM(α) indicates the hemispheric lateralization modulation of alpha power and $LV_{Th}$ , $LV_{CN}$ , $LV_{GP}$ refer to the lateralization volumes of thalamus, caudate nucleus and globus pallidus, respectively.

The analysis showed that the participants with larger volumes of the caudate nucleus in the left compared to the right hemisphere showed higher modulations in alpha power over the left compared to the right hemisphere (and vice versa). There was a trend for the same effect for the globus pallidus whereas the thalamus shows the opposite effect. These results were observed from the winning model that contained $LV_{Th}$, (β = –2.19, $T_{(29)}$ = –2.74, se = 0.80, p=0.010), $LV_{CN}$ (β = 0.92, $T_{(29)}$ = 2.83, se = 0.33, p-value = 0.008) and $LV_{GP}$ (β = 0.51, $T_{(29)}$ = 1.95, se = 0.26, p-value = 0.061) as regressors. This model predicted the HLM(α) values significantly in the GLM ($F_{3,29}$ = 7.4824, p=0.0007, adjusted $R^2$=0.376) as

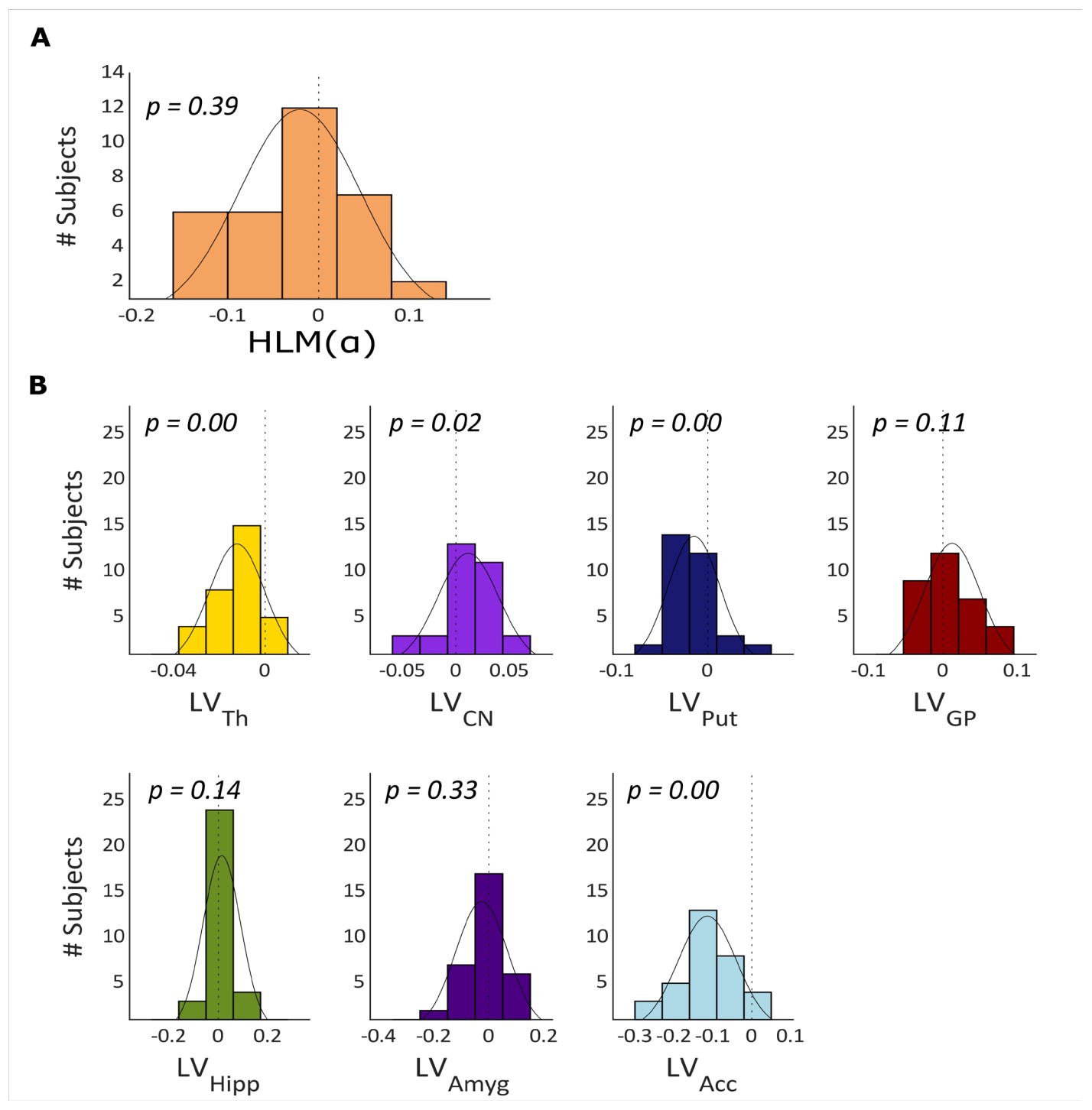

**Figure 3.** Hemispheric lateralization modulation (HLM(α)) grand average and basal ganglia volumes across all participants. (**A**) The HLM(α) distribution across participants. While there was considerable variation across participants, we observed no hemispheric bias in lateralized modulation values across participants (p-value = 0.39). (**B**) Histograms of the lateralization volumes of subcortical regions. We found that caudate nucleus was right lateralized (p-value = 0.021), whereas, putamen, nucleus accumbens, and thalamus volumes showed left lateralization (p-value = 0.004, p-value <0.001 and p-value <0.001, respectively). Th = Thalamus, CN = Caudate nucleus, Put = Putamen, GP = Globus pallidus, Hipp = Hippocampus, Amyg = Amygdala, Acc = Nucleus accumbens.

compared with an intercept-only null model (*Figure 4A*). These findings are illustrated in *Figure 4B*, confirming that both thalamus and caudate nucleus showed a significant linear partial regression with hemispheric lateralization modulation in the alpha band in the opposite and same direction. Although, the β estimate of LV$_{GP}$ only showed a positive trend, removing it from the regression resulted in worse models (*Supplementary files 1 and 2*).

It is worth noting that neither the behavioural nor the rapid invisible frequency tagging (RIFT) measures showed significant relationships with LVs and HLM(α) (*Figure 4—figure supplement 1* and *Supplementary file 3*).

### Association between volumetric lateralization of subcortical regions and attention related to perceptual load conditions

To relate load and salience conditions of the task to the relationship between subcortical structures and the alpha activity, we combined low-load or high-load targets with high-saliency or low-saliency distractors to manipulate the perceptual load appointed to each trial (Method section, *Figure 1*).

We therefore applied a multivariate multiple regression (MMR) using the HLM($\alpha$) values from each load/salience condition, and the LV values of the thalamus, caudate nucleus, and globus pallidus (*Equation 5*). Comparison of the full (i.e. MMR including the LV values of all seven subcortical structures as regressors) and reduced (i.e. MMR with all structures excluding the selected structures) models showed that our selected regressors predicted variability in HLM(α) values to an extent that was greater than chance ($F_{(25,28)} = 2.03$, p-value = 0.037). This was further confirmed when we compared the MMR model with the null model (i.e. MMR including only subject intercepts as regressor; $F_{(29,31)} = 3.78$, p-value = 0.0015). We next examined the extent to which LV values from each subcortical region predicted HLM(α) values for each load/salience condition. Our analysis, as shown in *Figure 5*, demonstrated that the thalamus had significant LV values in condition 1 (i.e. low-load target, non-salient distractor) with β = –3.63 ($T_{(29)} = -2.64$, se = 1.37, p-value = 0.0132). Globus pallidus showed a significant β coefficient in conditions 2 (i.e., high-load target, non-salient distractor) and 3 (i.e. low-load target, salient distractor) with β = 0.93, ($T_{(29)} = 2.15$, se = 0.43, p-value = 0.040) and β = 0.89 ($T_{(29)} = 2.30$, se = 0.39, p-value = 0.029), respectively. Condition 4 (i.e. high-load target, salient distractor) was the only condition in which the caudate nucleus had a β estimate significantly different than zero (β = 1.64, $T_{(29)} = 2.07$, se = 0.79, p-value = 0.049). To ascertain whether each predictor contributes to all conditions, we conducted statistical tests on the results of our MMR, testing the hypothesis that a given regressor does not impact all dependent variables. We found that while, with marginal significance, caudate nucleus can predict variability across all four of the task conditions ($F_{(26,4)} = 2.82$, p-value = 0.046), the predictive relationships of thalamus ($F_{(26,4)} = 2.43$, p-value = 0.073) with condition 1, and globus pallidus ($F_{(26,4)} = 2.29$, p-value = 0.087) with conditions 2 and 3 hold only for these conditions. In sum, this demonstrates that when the task is easiest (condition 1), the thalamus is related to alpha modulation. When the task is most difficult (condition 4), the caudate nucleus relates to the alpha modulation; however, its contributions are substantial enough to predict outcomes across all conditions. For the conditions with medium difficulty (conditions 2 and 3), the globus pallidus is related to the alpha band modulation.

## Discussion

In the current study, we sought to identify the association between the volumetric hemispheric asymmetries in subcortical structures and the hemispheric laterality in the modulation of posterior alpha oscillations during varying conditions of perceptual load. This association was tested in the context of a spatial attention paradigm where target load and distractor salience were manipulated. Our study resulted in two main findings: (1) globus pallidus, caudate nucleus, and thalamus predicted attention-related modulations of posterior alpha oscillations. (2) Each of these subcortical structures contributed differently to the lateralization values associated with the perceptual load conditions. For the easier task condition, the thalamus showed strong predictive power for alpha power modulation, whereas for mid-levels of load and salience, the globus pallidus showed predictive value. For the most perceptual demanding condition, we found that asymmetry of the caudate nucleus predicted alpha power modulation. These results shed light on the role of subcortical structures and their involvement in the modulation of oscillatory activity during the allocation of spatial attention.

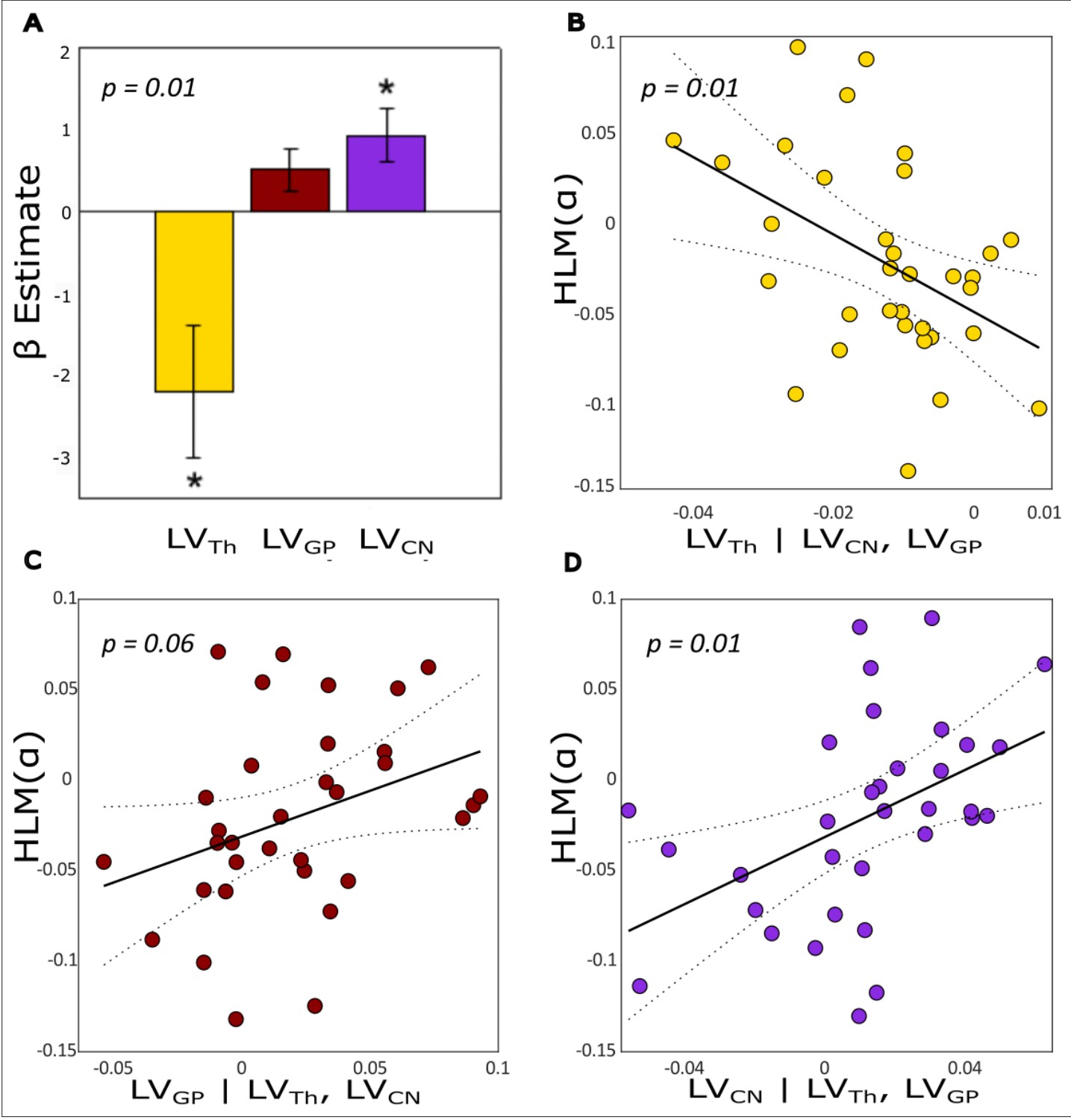

**Figure 4.** Lateralization volume of thalamus, caudate nucleus, and globus pallidus in relation to hemispheric lateralization modulation of alpha (HLM(α)) in the task. (**A**) The β coefficients for the best model (containing three regressors) associated with a generalized linear model (GLM) where lateralization volume (LV) values were defined as explanatory variables for HLM(α). The model significantly explained the HLM(α) (p-value = 0.0007). Error bars indicate standard errors of mean (SEM), n = 33. Asterisks denote statistical significance; *p-value <0.05. (**B**) Partial regression plot showing the association between $LV_{Th}$ and HLM(α) while controlling for $LV_{GP}$ and $LV_{CN}$ (p-value = 0.01). (**C**) Partial regression plot showing the association between $LV_{GP}$ and HLM(α) while controlling for $LV_{Th}$ and $LV_{CN}$ (p-value = 0.061). (**D**) Partial regression plot showing the association between $LV_{CN}$ and HLM(α) while controlling for $LV_{Th}$ and $LV_{GP}$ (p-value = 0.008). Negative (or positive) LVs indices denote greater left (or right) volume for a given substructure; similarly

*Figure 4 continued on next page*

*Figure 4 continued*

negative HLM(α) values indicate stronger modulation of alpha power in the left compared with the right hemisphere, and vice versa. The dotted curves in B, C, and D indicate 95% confidence bounds for the regression line fitted on the plot in red.

The online version of this article includes the following figure supplement(s) for figure 4:

**Figure supplement 1.** Lateralization volume of thalamus, caudate nucleus, and globus pallidus in relation to hemispheric lateralization modulation of rapid invisible frequency tagging (HLM(RIFT)) on the right and behavioural asymmetry on the left.

## Thalamus, caudate nucleus, and globus pallidus are involved in the allocation of spatial attention

While some MEG studies have demonstrated that it is possible to detect activity from deep structures such as the hippocampus (*Alberto et al., 2021*; *Griffiths et al., 2021*; *Meyer et al., 2017*; *Ruzich et al., 2019*), it is questionable whether one in general can use MEG to reliably detect activity from the thalamus and basal ganglia, owing to low SNR from sources close to the centre of the head (*Baillet, 2017*). Given these constraints, we instead correlated MEG data with structural magnetic resonance images to uncover functional contributions of subcortical structures to spatial attention.

We evaluated the relationship between subcortical structures and cortical oscillatory activity relying on the association between structure and function. Previous research points to a link between the volume of a given brain region and its functionality. It has been demonstrated that extensive navigation experience enlarges the size of right hippocampus (*Maguire et al., 2000*). Furthermore, in terms of neurological disorders, it is well established that shrinkage (atrophy) in specific regions is a predictor of a number of neurological and psychiatric conditions including Parkinson's disease, dementia, and Huntington's disease. In Parkinson's disease, atrophy in the nucleus accumbens and thalamus correlated with cognitive impairments (*Mak et al., 2014*). In a large-scale study on 773 participants, patients with Alzheimer's Disease have been shown to have a significantly smaller amygdala, thalamus, caudate nucleus, putamen, and nucleus accumbens than matched controls (*Yi et al., 2016*). It has also been shown that the spatial extent of pathological change in subcortical structures can predict cognitive changes in Parkinson's Disease (*Ye et al., 2022*). Patients with symptomatic Huntington's Disease also show significantly smaller caudate nucleus than pre-symptomatic participants who were carriers of Huntington's Disease gene mutation (*Aylward et al., 2000*). Changes in neocortical oscillatory activity have also been observed in neurological disorders which mainly are known to affect subcortical structures. For example, individuals with Alzheimer's Disease demonstrate an increase in slow oscillatory activities and a decrease in higher frequency oscillations (*Jafari et al., 2020*). Moreover, in patients with Parkinson's Disease, the power of beta oscillations increases relatively to when they are dopamine-depleted compared with when they are on dopaminergic medication (*Jenkinson and Brown, 2011*).

Based on these considerations, we argue that the volume of basal ganglia relates to the ability to modulate posterior brain oscillations in attention type tasks. We demonstrated this by considering the hemispheric lateralization of the basal ganglia structures in relation to the ability to modulate posterior alpha oscillations. Employing hemispheric lateralization was motivated by the organizational characteristic of structural asymmetry in healthy brain (*Kong et al., 2018*; *Guadalupe et al., 2017*). Additionally, considering the effects of aging (*Minkova et al., 2017*; *Guadalupe et al., 2017*) and neurodegenerative disorders, such as Alzheimer's Disease (*Roe et al., 2021*), on brain symmetry influenced this approach. Furthermore, computing lateralization indices for individuals addresses the challenge of accommodating variations in both head size and the power of oscillatory activity. Our findings are consistent with previous studies suggesting that thalamic and basal ganglia structures are involved in modulating oscillatory activity in the alpha band. For example, the largest nucleus of the thalamus, the pulvinar, supports the allocation of spatial attention by driving the oscillatory synchrony in the alpha band between cortical areas in a task-dependent manner (*Saalmann et al., 2012*) Also, our finding are consistent with other studies suggestions a role for the caudate nucleus (*Bogadhi et al., 2018*) and the pulvinar when allocating spatial attention (*Bogadhi et al., 2018*; *Green et al., 2017*; *Petersen et al., 1987*). Stimulation of the subthalamic nucleus has been shown to suppress oscillatory activity in the alpha and beta (8–22 Hz) frequency bands (*Abbasi et al., 2018*). Furthermore, our regression analysis outcomes align with the findings of *Mazzetti et al., 2019* underscoring the significant predictive influence exerted by the lateralized volume of globus pallidus on

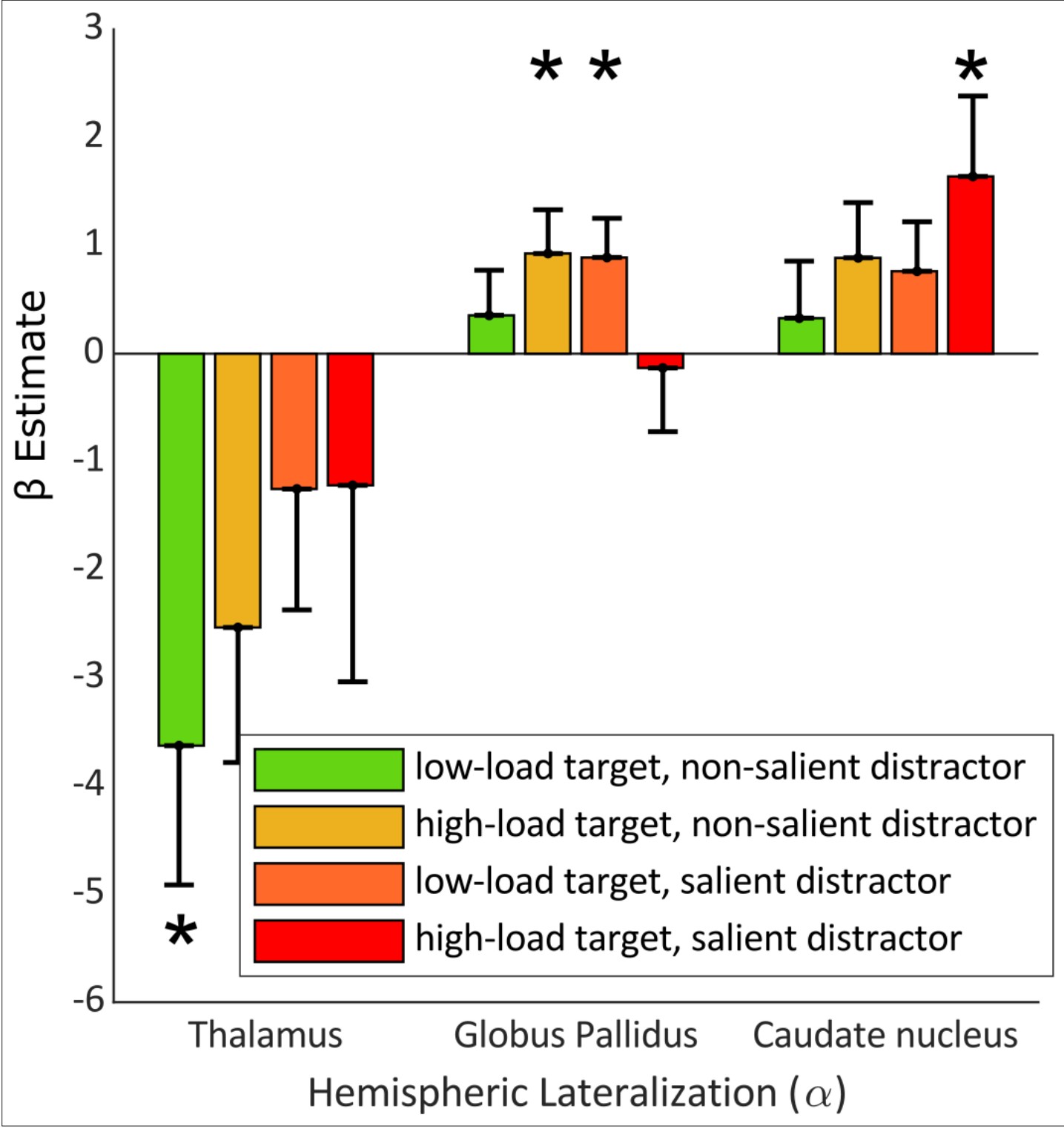

**Figure 5.** β estimates of subcortical nuclei from a multivariate regression model predicting HLM(α) in the four perceptual load conditions. Here, the HLM(α) values for the four load conditions are the dependent variables and the lateralization volume of subcortical structures are the explanatory variables. The model significantly explains HLM(α) variability (p-value = 0.001) in comparison with null model. Error bars indicate SEM, n = 33. Asterisks denote statistical significance; *p-value <0.05.

the modulation of hemispheric lateralization in alpha oscillations during spatial attention tasks. This convergence of results not only corroborates the validity and consistency of our findings but also extends the empirical foundation supporting the predictive role of the asymmetry of globus pallidus in modulating alpha oscillations beyond reward valence and to the context of attention.

## Thalamus, globus pallidus, and caudate nucleus play varying roles across different load conditions

Our results demonstrate a shift in the contribution of the thalamus, globus pallidus, and caudate nucleus when increasing the perceptual load of the target and saliency of the distractor. While in the low load, low saliency condition, the lateralized volume of the thalamus was correlated with the inter-hemispheric bias in alpha modulation, in the low load, high saliency, as well as high load, low saliency conditions, globus pallidus was related to the alpha oscillatory activity. Finally, the caudate nucleus was mainly associated with the high load, high saliency condition.

This differing pattern of the thalamic and basal ganglia structures might be suggestive of their respective contributions to the control of attentional resources. Involvement of the thalamus when the task is in its simplest form can be explained by its role relaying information between the basal ganglia and the prefrontal cortex (*McFarland and Haber, 2002*; *Jeon et al., 2014*). The opposite effect of the globus pallidus compared to the thalamus is striking, and possibly explained by the globus pallidus containing GABAergic interneurons. Thus the inhibitory nature of the globus pallidus projections to thalamus could explain why they are related to the alpha modulation in different manners (*Lanciego et al., 2012*). The involvement of the caudate nucleus in the most difficult condition is also in line with previous findings showing activation of caudate nucleus only in the higher level cognitive hierarchy in a working memory selection (*Chatham et al., 2014*) as well as a language task (*Jeon et al., 2014*). The engagement of globus pallidus might be reflected from its central role in harmonizing firing rates across the cortico-basal-ganglia circuits (*Crompe et al., 2020*). Globus pallidus also has wide projections to the thalamus (*Goldberg et al., 2013*) and can thereby impact the dorsal attentional networks by modulating prefrontal activities (*Nakajima et al., 2019*). Although these findings highlight the varying contributions of different regions, they do not imply a lack of evidence for correlations between these subcortical structures and other load conditions.

## Limitations and future directions

In the current study, we correlated the volumetric asymmetry of subcortical structures with the lateralized power of alpha oscillation. While this method provides novel insights into the role of subcortical structures in the modulation of oscillatory activity, it is indirect. The association between the function of subcortical nuclei and cortical oscillatory activity needs to be investigated further in electrophysiological studies that record the activity of both regions simultaneously. This could be done in non-human primates or in humans implanted with electrodes in the globus pallidus in treatment for Parkinson's Disease. In particular, EEG paired with globus pallidus recordings in participants performing spatial attention tasks would be of great value. Furthermore, in this study, our emphasis has been on assessing the size of subcortical structures. Future investigations could explore subcortical white matter connectivities and hemispheric asymmetries. This approach has previously been conducted on superior longitudinal fasciculus (SLF; *Marshall et al., 2015*; *D'Andrea et al., 2019*) and holds potential for examining cortico-subcortical connectivity in the context of oscillatory asymmetries.

Moreover, the current study faced methodological constraints, limiting the analysis to the entire thalamus. Additionally, we refrained from directly comparing the contributions of subcortical structures to different conditions due to low statistical power. It would be of great interest to conduct further investigations to quantify the distinct impacts of individual thalamic nuclei on the association between subcortical structures and the modulation of oscillatory activity. In future studies, it would be interesting to design an experiment directly addressing which subcortical regions contribute to distractor and target load in terms of modulating the alpha band activity. In order to ensure sufficient statistical power for doing so possibly each factor needs to be addressed in different experiments.

Moreover, our failure to identify a relationship between the lateralized volume of subcortical structures and behavioural measures should be addressed in studies that are better designed to capture performance asymmetries (*Ghafari et al., 2023*). Individual preferences toward one hemifield, which were not addressed in the current study design, could potentially strengthen the power to detect

correlations between structural variations in the subcortical structures and behavioural measures. For example, it would be of great significance to investigate the lateralization of subcortical structures in patients with hemineglect in relation to right hemisphere lesions (**Buxbaum et al., 2004**).

We did not find any association between the power of RIFT signal and the size asymmetry of subcortical structures. Since Bayes factors were less than 0.1, we conclude that our RIFT null findings are robust, suggesting a dissociation between how alpha oscillations and neuronal excitability indexed by RIFT relate to subcortical structures. In previous work we have demonstrated that the attention modulation of the RIFT signal is strongest observed in early visual cortex, whereas alpha oscillations are more strongly modulated around the parieto-occipital sulcus (**Zhigalov et al., 2019**). It has been proposed that the modulation in RIFT in early visual cortex with attention reflects gain control. According to this framework, we conclude that subcortical regions might not be involved in gain modulation in early visual cortex during the allocation of spatial attention, but rather in the downstream gating of visual information.

## Conclusion

Our findings point to a link between thalamus and nuclei of the basal ganglia and measures of alpha oscillations in relation to spatial attention. Moreover, they demonstrate distinguished contributions of the different subcortical structures depending on target load or distractor salience, thus informing theories of how subcortical structures relate to oscillatory dynamics in challenging attentional settings. The stage is now set for further investigating the relationship between subcortical regions and the modulation of oscillatory activity. Linking brain oscillations to changes in subcortical regions associated with neurological disorders, such as Alzheimer's Disease (**Yi et al., 2016**; **Jiji et al., 2013**) and Parkinson's Disease (**Mak et al., 2014**), could have potential clinical applications in terms of early diagnosis. Our approach could also be extended to other tasks resulting in hemispheric lateralization of oscillatory brain activity, for example working memory tasks (**Sauseng et al., 2009**) or language tasks (**Wang et al., 2013**). Our results also call for more direct investigations of the relationship between subcortical regions and neocortical oscillations which is best done by intracranial recordings in non-human primates or utilizing human recording from deep-brain stimulation electrodes combined with EEG or MEG.

## Materials and methods

### Participants

We analysed a previously collected dataset, described in **Gutteling et al., 2022**. 35 right-handed healthy volunteers (25 female, mean age: 24 ± 5.7) participated. All reported normal or corrected-to-normal vision. One participant did not give consent for their data to be used outside of the original study and one was removed due to poor MRI [segmentation] quality, resulting in 33 participants in total. All subjects signed an informed consent form before participation and were paid £15 per hour. The study was conducted in compliance with the Declaration of Helsinki and was approved by the Science, Technology, Engineering, and Mathematics (STEM) ethical review committee of the University of Birmingham (ERN_18-0226AP4).

### Experimental design

Participants were instructed to perform a cued change detection task (2 blocks of 256 trials, 45 min; *Figure 1A*), designed to assess selective attention function under varying conditions of perceptual challenge. Each trial started with a fixation point (1000ms) followed by two faces presented on the left and right side of the screen (1000ms). The fixation cross then turned into an arrowhead for 350ms cueing the left or the right hemifield. After a variable 1000–2000ms delay, the eye-gaze of each face randomly shifted rightward or leftward in a 150ms interval. Then followed a 1000ms response interval where participants were asked to respond with their right or left index finger whether the gaze direction of the cued face shifted left or right (NAtA technologies, Coquitlam, BC, Canada). The experimental paradigm was implemented on a Windows 10 computer running MATLAB (Mathworks Inc, Natrick, USA) using Psychophysics Toolbox 3.0.11 (**Brainard, 1997**; **Pelli, 1997**).

### Visual stimuli

Stimuli were circular faces that comprised 8° visual angle in diameter and placed with 7° eccentricity from fixation and were presented in the lower hemifield. Over trials, the perceptual load of targets

was manipulated using a noise mask; noisy targets are harder to detect than clear targets and therefore incur greater perceptual load in their detection. The saliency of distractor stimuli was also manipulated using a noise mask; noisy distractor stimuli are less salient than clear distractors and therefore less disruptive to performance on the detection task. The noise mask was created by randomly swapping 50% of the stimulus pixels (*Figure 1B*). This manipulation resulted in four target-load/distractor-saliency conditions: (*Nobre and Kastner, 2014*) target: low load, distractor: low saliency (i.e. clear target, noisy distractor), (*Desimone and Duncan, 1995*) target: high load, distractor: low saliency (i.e. noisy target, noisy distractor), (*Moran and Desimone, 1985*) target: low load, distractor: high saliency (i.e. clear target, clear distractor), (*Corbetta and Shulman, 2002*) target: high load, distractor: high saliency (i.e. noisy target, clear distractor) (*Figure 1B and C*). The stimulus set consisted of eight different face identities that were randomized across trials. On each trial, the identities of both stimuli were the same; however, to avoid visual differences between left and right the faces were mirror symmetric from the fixation point. Stimuli were projected using a VPixx PROPixx projector (VPixx technologies, Saint-Bruno, Canada) in Quad RGB mode (1440 Hz) with an effective resolution of 960x540 pixels. Face stimuli were tagged with an invisible rapid-frequency-tagged flicker (for more details please refer to *Gutteling et al., 2022*). The distance between the participant and the projection screen was 148 cm resulting in a 25.6° of visual angle screen.

## Structural data acquisition

T1-weighted magnetic resonance images were acquired for 10 participants on a 3 Tesla Magnetom Prisma whole-body scanner (Siemens AG) with acquisition parameters: TR/TE = 2000/2.01ms, TI = 880ms, FoV = 256 × 256×208 mm$^3$, acquired voxel size = 1 × 1 x1 mm$^3$. For 23 participants MRI images were attained from previous studies. These scans were obtained at the former Birmingham University Imaging Center (3-Tesla Philips Achieva Scanner: TR/TE = 7.4/3.5ms, FA = 7°, FOV = 256 × 256 x176 mm$^3$, acquired voxel size = 1 × 1 x1 mm$^3$) were used. The two remaining participants provided their MRIs from other sources.

## Structural data analysis

To segment the subcortical structures, FMRIB's Integrated Registration and Segmentation Tool (FIRST) v5.0.9 (https://www.fmrib.ox.ac.uk/fsl/, Oxford Centre for Functional MRI of the Brain) was used. FIRST is an automated model-based tool that runs a two-stage affine transformation to MNI152 space, to achieve a robust pre-alignment of thalamus, caudate nucleus, putamen, globus pallidus, hippocampus, amygdala, and nucleus accumbens based on individual's T1-weighted MR images. Subcortical structures are modelled within a Bayesian framework (using manually segmented images provided by the Centre for Morphometric Analysis, CMA, MGH, Boston, as a prior) as surface meshes (masks) that were then fit to the registered image. Regions outside of the masks were excluded from subcortical alignment (*Patenaude et al., 2011*).

To assess hemispheric laterality for each subcortical structure, we calculated the Lateralization Volume indices (LVs):

$$LV_s = \frac{V_{s_{right}} - V_{s_{left}}}{V_{s_{right}} + V_{s_{left}}} \tag{1}$$

where $V_{s_{right}}$ and $V_{s_{left}}$ represent the anatomical volume of a given subcortical structure (s) in number of voxels, in the right and left hemisphere, respectively. This equation implicitly controls for individual differences in brain volumes and has been commonly used to compute hemispheric structural asymmetries (*Mazzetti et al., 2019*). LVs can range between –1 and 1 where a positive LV indicates rightward asymmetry and vice versa.

## MEG data acquisition

Electromagnetic data were recorded from participants while seated in upright position, using a 306-sensor whole-head TRIUX system from MEGIN (MEGIN, Stockholm, Sweden) including 102 magnetometers and 204 (2x102 orthogonal) planar gradiometers. The MEG data were sampled at 1000 Hz, following an embedded anti-aliasing low-pass filter at 330 Hz and stored for offline analysis. Head position of the participants was monitored by coils placed on anatomical fiducials (nasion, left and right periauricular points), digitized using a Polhemus Fastrack electromagnetic digitizer system

(Polhemus Inc). Eye movements were recorded using an Eyelink eyetracker (EyeLink 1000, SR research Ltd., Ottawa, Canada) along with vertical EOG sensors.

## MEG data analysis

MEG data analysis was performed using custom scripts in MATLAB 2017a and 2019b (The Math-Works) and the FieldTrip toolbox (*Oostenveld et al., 2011*). The analysis pipeline was adapted from the FLUX pipeline (*Ferrante et al., 2022*) and the scripts are available on GitHub (copy archived at *Ghafari, 2024*).

### Preprocessing

Raw MEG data were high-pass filtered at 1 Hz and demeaned. Then data were segmented in 4 s epochs (–3s to 1s) relative to the target-onset (gaze shift of the face stimuli). Secondly, trials with sensors artifacts (e.g. jumps) were removed manually to prepare the data for automatic artifact attenuation using independent component analysis (ICA; '*runica.m*' in FieldTrip). Components related to eye blinks/movements, heartbeat and muscle activity were rejected. Thirdly, by visually inspecting the trials, we removed those containing clear residual artifacts such as eye blinks. We also removed trials with saccadic deviations larger than 3° from fixation (using EyeLink eye tracker data) during the 1.5 s interval before target-onset (–1.5–0 s; average ± SD = 13.7%±8.0 trials). Sensors that were removed during preprocessing were interpolated using a weighted neighbor estimate.

### Time-frequency analysis of power

To calculate the time frequency representations (TFR) of power, we used a 3-cycle fixed time-window (e.g. 300ms for 10 Hz) at each 10ms step. The data segments were multiplied by a Hanning taper to control the frequency smoothing and reduce spectral leakage. For computational efficiency, we also used a zero-padding, rounding up the length of segments to the next power of 2. Then a fast Fourier transform (FFT) was applied to the tapered segments in the 2–30 Hz frequency range in 1 Hz steps and the power was estimated. The power was then summed for each gradiometer pair.

To quantify the anticipatory oscillatory activity, we focussed on the –850–0ms interval before target onset. To select sensors constituting the region of interest (ROI), we calculated the 8–13 Hz alpha modulation index (MI($\alpha$)) for all sensors. TFR of power for each sensor was averaged over all trials in the –850–0ms interval, for attention to right and left. Then the MI($\alpha$) for each participant and each sensor was calculated as:

$$MI\left(\alpha\right)_k = \frac{Power\left(\alpha\right)_{k_{att\ right}} - Power\left(\alpha\right)_{k_{att\ left}}}{Power\left(\alpha\right)_{k_{att\ right}} + Power\left(\alpha\right)_{k_{att\ left}}} \tag{2}$$

where Power($\alpha$)$_k$ denotes the alpha power at sensor *k* in each condition.

Subsequently, at the group level, MI($\alpha$) for all sensors on the left hemisphere were subtracted from the corresponding sensors on the right hemisphere. The resulting values were then sorted and five pairs of sensors ($n_{ROI}$) that showed the highest difference in MI($\alpha$) values were selected, resulting in 10 sensors, symmetrically distributed over the right and left hemispheres. Ten sensors were selected to ensure sufficient coverage of the region exhibiting alpha modulation as judged from prior work (*Zhigalov et al., 2019*). As MI($\alpha$) consistently represents power of alpha in attention right versus attention left conditions, it entails the comparison between ipsilateral and contralateral alpha modulation power for sensors located on the right side of the head. The same comparison applies inversely for sensors situated on the left side of the brain.

To evaluate hemisphere-specific lateralization of alpha band modulation, we applied the hemispheric lateralization modulation (HLM($\alpha$)) index:

$$HLM\left(\alpha\right) = \frac{1}{n_{ROI}} \sum_{k=1}^{n_{ROI}} MI\left(\alpha\right)_{k_{right}} + \frac{1}{n_{ROI}} \sum_{k=1}^{n_{ROI}} MI\left(\alpha\right)_{k_{left}} \tag{3}$$

where $n_{ROI}$ = 5 represents the number of sensors in each ROI and $MI\left(\alpha\right)_{k_{right}}$ or $MI\left(\alpha\right)_{k_{left}}$ denote the modulation index for sensor *k* over the right or left hemisphere, respectively.

We computed the modulation index (MI) for rapid invisible frequency tagging (RIFT) by averaging the power of the signal in sensors on the right when attention was directed to the right compared to

when it was directed to the left. This calculation was also performed for sensors on the left. Consequently, we identified the top 5 sensors on each side with the highest MI as the Region of Interest (ROI). Utilizing the sensors within the ROI, we computed hemispheric lateralization modulation (HLM) of RIFT by summing the average MI(RIFT) of the right sensors and the average MI(RIFT) of the left sensors, obtaining one HLM(RIFT) value for each participant. For a more comprehensive analysis, refer to reference (*Gutteling et al., 2022*).

## Statistical analysis

### Generalized linear model

To model how the mean expected value of HLM($\alpha$) indices depends on the lateralized volume of subcortical structures, we applied a generalized linear model (GLM) using HLM($\alpha$) values as the dependent variable and LV indices of subcortical structures as the systematic (explanatory) variables. We performed a collinearity analysis (*vif.m* function in MATLAB) to ensure that the predictor variables were sufficiently independent prior to performing the GLM analysis.

First, we sought to determine the best set of regressors that predicted variability in HLM($\alpha$) values. We therefore used all possible combinations of regressors (LVs; one to seven combinations) in a linear mixed-effects model (*fitme.m* function in MATLAB) to predict HLM($\alpha$) indices and selected the model that scored the lowest using the Akaike information criterion (AIC; *Akaike, 1974*) score as the winning model. We confirmed our findings using Bayesian information criterion (BIC; *Schwarz, 1978*) and produced similar results. These values are commonly used to identify the best point of trade-off between fit and model complexity.

To estimate the $\beta$ weights of the winning model the optimal set of regressors (here $LV_{Th}$, $LV_{CN}$, and $LV_{GP}$) were used as the explanatory variables in a GLM (*fitlm.m* function in MATLAB) to predict HLM(a) values with the following formula:

$$HLM\left(\alpha\right) \sim \beta_0 + \beta_1 LV_{Th} + \beta_2 LV_{CN} + \beta_3 LV_{GP} + \varepsilon \tag{4}$$

Here, $LV_{Th}$, $LV_{CN}$, and $LV_{GP}$ refer to the lateralization volume of thalamus, caudate nucleus, and globus pallidus, respectively.

The absence of a relationship between modulations of alpha oscillations and the hippocampus and amygdala was expected as these regions typically are not associated with the allocation of spatial attention and thus add validity to our approach.

### Multivariate multiple regression

To simultaneously model the predictive relationship between the lateralized volume of thalamus, caudate, and globus pallidus, and all four load conditions, we used a multivariate multiple regression (MMR) (*Manly and Navarro Alberto, 2016*) analysis. MMR is used to predict multiple dependent variables using multiple systematic parameters. It allows for modifying our hypothesis tests and confidence intervals for explanatory parameters and responses, respectively (*Dattalo, 2013*). The model was defined as:

$$HLM\left(\alpha_1\right) + HLM\left(\alpha_2\right) + HLM\left(\alpha_3\right) + HLM\left(\alpha_4\right) \sim \beta_0 + \beta_1 LV_{Th} + \beta_2 LV_{CN} + \beta_3 LV_{GP} + \varepsilon \tag{5}$$

Where HLM($\alpha$) refers to hemispheric lateralization modulation of alpha power in load conditions 1–4 (*Figure 1C*), respectively; $\beta$ refers to the coefficients in the model; $LV_{Th}$, $LV_{CN}$, and $LV_{GP}$ refer to the lateralization volume of thalamus, caudate nucleus, and globus pallidus, respectively.

To ensure our chosen MMR predicts meaningful variance in HLM($\alpha$) scores, we compared a full model containing LV indexes from all 7 subcortical regions to one where the key structures of interest (i.e. thalamus, caudate nucleus, and globus pallidus) had been removed, leaving putamen, nucleus accumbens, hippocampus, and amygdala as regressors. This model is referred to as the reduced model. We also compared a model containing the key regressors of interest ($LV_{Th}$, $LV_{CN}$, $LV_{GP}$) to a null model that contained only subject intercepts as regressors. Models were compared one-way ANOVA test in RStudio (version 2022.02.0; *R Development Core Team, 2020*).

To examine the specificity of each regressor for lateralized alpha in each condition, we statistically assessed the results of the MMR against the null hypothesis that a particular predictor does not

contribute to all dependent variables, employing a MANOVA test in RStudio (version 2022.02.2; *R Development Core Team, 2020*).

### Behavioral data analysis

To evaluate if the participants response times and accuracy was correlated with the hemispheric lateralization of alpha oscillatory activity as well as lateralized volume of subcortical structures, we calculated behavioral asymmetry (BA) as below:

$$BA_{ACC/RT} = \frac{ACC/RT_{att\ right} - ACC/RT_{att\ left}}{ACC/RT_{att\ right} + ACC/RT_{att\ left}} \tag{6}$$

where $ACC/RT_{att\ right}$ and $ACC/RT_{att\ left}$ correspond to the behavioural asymmetric performance in accuracy or response times when the attention was toward right or left visual hemifield, respectively. Finally, we ran the winning GLM model with accuracy and response times as the dependent variable and $LV_{Th}$, $LV_{CN}$, and $LV_{GP}$ as the regressors.

## Acknowledgements

This work was funded by the Ministry of Defence, UK, through the mTBI Predict Consortium and supported by the NIHR Oxford Health Biomedical Research Centre (NIHR203316) and a Wellcome Trust Discovery Award (grant number 227420). The views expressed are those of the author(s) and not necessarily those of the NIHR or the Department of Health and Social Care. The funders had no role in the preparation of the manuscript or decision to publish. The computations described in this paper were performed using the University of Birmingham's BlueBEAR HPC service, which provides a High Performance Computing service to the University's research community. See http://www.birmingham.ac.uk/bear for more details. We express our gratitude to Jonathan L Winter for assisting us with the MEG recordings.

## Additional information

### Funding

| Funder | Grant reference number | Author |
| --- | --- | --- |
| Ministry of Defence | | Ole Jensen |
| NIHR Oxford Biomedical Research Centre | NIHR203316 | Ole Jensen |
| Wellcome Trust | 227420 | Ole Jensen |

The funders had no role in study design, data collection and interpretation, or the decision to submit the work for publication. For the purpose of Open Access, the authors have applied a CC BY public copyright license to any Author Accepted Manuscript version arising from this submission.

### Author contributions

Tara Ghafari, Conceptualization, Data curation, Software, Formal analysis, Validation, Investigation, Visualization, Methodology, Writing - original draft, Writing - review and editing; Cecilia Mazzetti, Conceptualization, Supervision, Visualization, Methodology; Kelly Garner, Tjerk Gutteling, Supervision, Methodology, Writing - review and editing; Ole Jensen, Conceptualization, Resources, Supervision, Funding acquisition, Validation, Methodology, Writing - review and editing

### Author ORCIDs

Tara Ghafari http://orcid.org/0000-0002-5178-8702
Kelly Garner http://orcid.org/0000-0002-0690-6941
Ole Jensen http://orcid.org/0000-0001-8193-8348

### Ethics

Human subjects: The study was conducted in compliance with the Declaration of Helsinki and was approved by the Science, Technology, Engineering, and Mathematics (STEM) ethical review committee of the University of Birmingham (ERN_18-0226AP4).

Reviewer #1 (Public Review): https://doi.org/10.7554/eLife.91650.3.sa1
Reviewer #2 (Public Review): https://doi.org/10.7554/eLife.91650.3.sa2
Author response https://doi.org/10.7554/eLife.91650.3.sa3

---

## Additional files

### Supplementary files

• Supplementary file 1. Akaike Information Criterion (AIC) and Bayesian Information Criterion (BIC) values for all possible combinations of regressors (Lateralized Volume of subcortical structures). The selected model, with lowest AIC, is marked in green.

• Supplementary file 2. The combination of structures for each regression model above. The selected model is marked in green.

• Supplementary file 3. Bayes factors for correlation between hemispheric laterality of subcortical structures with hemispheric lateralization modulation of rapid invisible frequency tagging (HLM(RIFT)) and with behavioural asymmetry (BA). The Pearson correlation between each subcortical structure with HLM(RIFT) and behavioural asymmetry was calculated. The likelihood of the data under the alternative hypothesis (the evidence of correlation) were subsequently compared to the likelihood under null hypothesis (absence of correlation), given the data. As it is demonstrated in the table, all Bayes factors were below or very close to 1 indicating evidence for the null hypothesis.

• MDAR checklist

### Data availability

All the codes to run the analysis for this work is available on GitHub (copy archived at *Ghafari, 2024*).

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
