## [Editor Report · eLife assessment]

This study by Ghafari et al. tackles a question relevant for the field of attention as it connects structural differences in subcortical regions with oscillatory modulations during attention allocation. Using a combination of Magnetoencephalography (MEG) and magnetic resonance imaging (MRI) data in human subjects, the **valuable** results show that inter-individual differences in the lateralisation of alpha oscillations are explained by asymmetry of subcortical brain regions. The strength of evidence is deemed **convincing** in line with current state-of-the-art.

---

## [Referee Report · Reviewer #1 (Public Review)]

Summary:

The authors re-analysed the data of a previous study in order to investigate the relation between asymmetries of subcortical brain structures and the hemispheric lateralization of alpha oscillations during visual spatial attention. The visual spatial attention task crossed the factors of target load and distractor salience, which made it possible to also test the specificity of the relation of subcortical asymmetries to lateralized alpha oscillations for specific attentional load conditions. Asymmetry of globus pallidus, caudate nucleus, and thalamus explained inter-individual differences in attentional alpha modulation in the left versus right hemisphere. Multivariate regression analysis revealed that the explanatory potential of these regions' asymmetries varies as a function of target load and distractor salience.

In the revision of the article, the authors addressed my concerns.

However, my concern with regard to the statistical analysis of the specificity of certain subcortical regions predicting HLM seems to be not fully addressed. The authors added an additional statistical analysis for "testing the null hypothesis that a given regressor does not impact all dependent variables". To my understanding, this is a somewhat unusual definition of a null hypothesis. Typically, the null hypothesis is the hypothesis of no effect, meaning here it should state that the effect is the same across predictors.

In the new statistical analysis, the authors seem to take non-significant results (p>.05) as evidence for the specificity of subcortical regions in predicting HLM. The rationale of this statistical approach is difficult to follow and was somewhat unclear to me.

A much simpler and more straight-forward approach would be to contrast beta-estimates per subcortical region between experimental conditions. For instance, if the beta estimates in the thalamus for the "low-load target, non-salient distractor" condition would be significantly larger than the beta estimates for the other conditions, this would speak to specificity.

---

## [Referee Report · Reviewer #2 (Public Review)]

Summary:

In this study, Ghafari et al. explored the correlation between hemispheric asymmetry in the volume of various subcortical regions and lateralization of posterior alpha band oscillations in a spatial attention task with varying cognitive demands. To this end, they combined structural MRI and task MEG to investigate the relationship between hemispheric differences in volume of basal ganglia, thalamus, hippocampus and amygdala and hemisphere-specific modulation of alpha-band power. The authors report that differences in the thalamus, caudate nucleus and globus pallidus volume are linked to the attention-related changes in alpha band oscillations with differential correlations for different regions in different conditions of the design (depending on the salience of the distractor and/or the target).

The manuscript contributes to filling an important gap in current research on attention allocation which commonly focuses exclusively on cortical structures. Because it is not possible to reliably measure subcortical activity with non-invasive electrophysiological methods, they correlate volumetric measurements of the relevant subcortical regions with cortical measurements of alpha band power. Specifically, they build on their own previous finding showing a correlation between hemispheric asymmetry of basal ganglia volumes and alpha lateralization by assessing a task without an explicit reward component. Furthermore, the authors use differences in saliency and perceptual load to disentangle the individual contributions of the subcortical regions. These remain somewhat hard to interpret, given their post hoc nature, and the lack of statistical power to compare task demand effects directly, but the results raise interesting new hypotheses for future work.

---

## [Author Response]

The following is the authors’ response to the original reviews.

**eLife assessment**
The study by Ghafari et al. addresses a question that is highly relevant for the field of attention as it connects structural differences in subcortical regions with oscillatory modulations during attention allocation. Using a combination of magnetoencephalography (MEG) and magnetic resonance imaging (MRI) data in human subjects, inter-individual differences in the lateralization of alpha oscillations are explained by asymmetry of subcortical brain regions. The results are important, and the strength of the evidence is convincing. Yet, clarifying the rationale, reporting the data in full, a more comprehensive analysis, and a more detailed discussion of the implications will strengthen the manuscript further.
**Public Reviews:**

**Reviewer #1 (Public Review):**
Summary:The authors re-analysed the data of a previous study in order to investigate the relation between asymmetries of subcortical brain structures and the hemispheric lateralization of alpha oscillations during visual spatial attention. The visual spatial attention task crossed the factors of target load and distractor salience, which made it possible to also test the specificity of the relation of subcortical asymmetries to lateralized alpha oscillations for specific attentional load conditions. Asymmetry of globus pallidus, caudate nucleus, and thalamus explained inter-individual differences in attentional alpha modulation in the left versus right hemisphere. Multivariate regression analysis revealed that the explanatory potential of these regions' asymmetries varies as a function of target load and distractor salience.Strengths:The analysis pipeline is straightforward and follows in large parts what the authors have previously used in Mazzetti et al (2019). The authors use an interesting study design, which allows for testing of effects specific to different dimensions of attentional load (target load/distractor salience). The results are largely convincing and in part replicate what has previously been shown. The article is well-written and easy to follow.

We thank the reviewer for their interest in our study.

Weaknesses:While the article is interesting to read for researchers studying alpha oscillations in spatial attention, I am somewhat sceptical about whether this article is of high interest to a broader readership. Although I read the article with interest, the conceptual advance made here can be considered mostly incremental. As the authors describe, the present study's main advance is that it does not include reward associations (as in previous work) and includes different levels of attentional load. While these design features and the obtained results indeed improve our general understanding of how asymmetries of subcortical structures relate to lateralized alpha oscillations, the conceptual advance is somewhat limited.

We thank the reviewer for their constructive comment. We’d like to highlight that this is the first study to show relationship between subcortical structures asymmetry with attention-modulated alpha oscillation that did not involve any reward-associations- which is the most studied role of basal ganglia. We also believe there is value is having a second study linking the asymmetry in volume of subcortical structures to the modulation of alpha oscillations as this surprising finding also have important clinical implications (see below). We edited the manuscript as below to explain the advances made in this study:

Introduction (Line 112): “Our current findings broaden our understanding of how subcortical structures are involved in modulating alpha oscillations during top-down spatial attention, in the absence of any reward or value associations. “

Discussion (Line 301): “It has also been shown that the spatial extent of pathological change in subcortical structures can predict cognitive changes in Parkinson’s Disease (43). […] Changes in neocortical oscillatory activity have also been observed in neurological disorders which mainly are known to affect subcortical structures. For example, individuals with Alzheimer's Disease demonstrate an increase in slow oscillatory activities and a decrease in higher frequency oscillations (45). Moreover, in patients with Parkinson’s Disease, the power of beta oscillations increases relatively to when they are dopamine-depleted compared with when they are on dopaminergic medication (46).”

While the analysis of the relation of individual subcortical structures to alpha lateralization in different attentional load conditions is interesting, I am not convinced that the present analysis is suited to draw strong conclusions about the subcortical regions' specificity. For example, the Thalamus (Fig. 5) shows a significant negative beta estimate only in one condition (low-load target, non-salient distractor) but not in the other conditions. However, the actual specificity of the relation of thalamus asymmetry to lateralized alpha oscillations would require that the beta estimate for this one condition is significantly higher than the beta estimates for the other three conditions, which has not been tested as far as I understand.

We thank the reviewer for this constructive comment. We agree with the reviewer that we should compare the β value amongst the conditions. We therefore determined to better harness the multivariate nature of our analysis. Multivariate regression analysis allows one to test the null hypothesis that a given predictor does not contribute to all the dependent variables. A rejection of this hypothesis would suggest that lateralization of a given region of interest significantly predicts variability across all 4 of the task conditions, whereas failure to reject the null would imply that the predictive relationship holds only for that single condition. We tested this global null hypothesis using a MANOVA test and found the following which we have added to the manuscript:

Results (Line 250): “To ascertain whether each predictor contributes to all conditions, we conducted statistical tests on the results of our MMR using the null hypothesis that a given regressor does not impact all dependent variables. We found that while, with marginal significancy, caudate nucleus can predict variability across all four of the task conditions (F(26,4) = 2.82, p-value = 0.046), the predictive relationships of thalamus (F(26,4) = 2.43, p-value = 0.073) with condition 1, and globus pallidus (F(26,4) = 2.29, p-value = 0.087) with conditions 2 and 3 hold only for these conditions. In sum, this demonstrates that when the task is easiest (condition 1), the thalamus is related to alpha modulation. When the task is most difficult (condition 4), the caudate nucleus relates to the alpha modulation, however, its contributions are substantial enough to predict outcomes across all conditions. For the conditions with medium difficulty (conditions 2 and 3) the globus pallidus is related to the alpha band modulation. “

Method (Line 599): “To examine the specificity of each regressor for lateralized alpha in each condition, we statistically assessed the results of the MMR against the null hypothesis that a particular predictor does not contribute to all dependent variables, employing a MANOVA test in RStudio (version 2022.02.2) (80).”

Discussion (Line 337): “Thalamus, Globus Pallidus, and Caudate nucleus play varying roles across different load conditions.”

Discussion (Line 361): “Although these findings highlight the varying contributions of different regions, they do not imply a lack of evidence for correlations between these subcortical structures and other load conditions.”

Discussion (Line 379): “Additionally, we refrained from directly comparing the contributions of subcortical structures to different conditions due to low statistical power. […] In future studies it would be interesting to design an experiment directly addressing which subcortical regions contribute to distractor and target load in terms of modulating the alpha band activity. In order to ensure sufficient statistical power for doing so possibly each factor needs to be addressed in different experiments.”

**Reviewer #3 (Public Review):**
Summary:In this study, Ghafari et al. explored the correlation between hemispheric asymmetry in the volume of various subcortical regions and lateralization of posterior alpha-band oscillations in a spatial attention task with varying cognitive demands. To this end, they combined structural MRI and task MEG to investigate the relationship between hemispheric differences in the volume of basal ganglia, thalamus, hippocampus, and amygdala and hemisphere-specific modulation of alpha-band power. The authors report that differences in the thalamus, caudate nucleus, and globus pallidus volume are linked to the attention-related changes in alpha band oscillations with differential correlations for different regions in different conditions of the design (depending on the salience of the distractor and/or the target).Strengths:The manuscript contributes to filling an important gap in current research on attention allocation which commonly focuses exclusively on cortical structures. Because it is not possible to reliably measure subcortical activity with non-invasive electrophysiological methods, they correlate volumetric measurements of the relevant subcortical regions with cortical measurements of alpha band power. Specifically, they build on their own previous finding showing a correlation between hemispheric asymmetry of basal ganglia volumes and alpha lateralization by assessing a task without an explicit reward component. Furthermore, the authors use differences in saliency and perceptual load to disentangle the individual contributions of the subcortical regions.

We appreciate the reviewer’s interest in our study.

Weaknesses:The theoretical bases of several aspects of the design and analyses remain unclear. Specifically, we missed statements in the introduction about why it is reasonable, from a theoretical perspective, to expect:(i) a link between volumetric measurements and task activity;

We thank the reviewer for this constructive feedback. We have now addressed this concern in the revised manuscript.

Discussion (Line 293): “It has been demonstrated that extensive navigation experience enlarges the size of right hippocampus (40). Furthermore, in terms of neurological disorders, it is well established that shrinkage (atrophy) in specific regions is a predictor of a number of neurological and psychiatric conditions including Parkinson’s disease, dementia, and Huntington’s disease. […] It has also been shown that the spatial extent of pathological change in subcortical structures can predict cognitive changes in Parkinson’s Disease (43). […] Changes in neocortical oscillatory activity have also been observed in neurological disorders which mainly are known to affect subcortical structures. For example, individuals with Alzheimer's Disease demonstrate an increase in slow oscillatory activities and a decrease in higher frequency oscillations (45). Moreover, in patients with Parkinson’s Disease, the power of beta oscillations increase relatively to when they are dopamine-depleted compared with when they are on dopaminergic medication (46). “

(ii) a specific link with hemispheric asymmetry in subcortical structures (While focusing on hemispheric lateralization might circumvent the problem of differences in head size, it would be better to justify this focus theoretically, which requires for example a short review of evidence showing ipsilateral vs contralateral connections between the relevant subcortical and cortical structures);

We thank the reviewer for this helpful comment that resulted in clarification of the manuscript. We addressed this issue in the revised manuscript; we also now have complemented the revised manuscript with papers directly investigating asymmetry of subcortical regions in relation to neurological disorders:

Introduction (Line 102): “We utilized the hemispheric laterality of subcortical structures and alpha modulation to overcome issues related to individual variations in oscillatory power and head size.”

Discussion (Line 314): “Employing hemispheric lateralization was motivated by the organizational characteristic of structural asymmetry in healthy brain (47). Additionally, considering the effects of aging (48) and neurodegenerative disorders, such as Alzheimer's Disease (49), on brain symmetry influenced this approach. Furthermore, computing lateralization indices for individuals addresses the challenge of accommodating variations in both head size and the power of oscillatory activity.”

Discussion (Line 374): “Furthermore, in this study, our emphasis has been on assessing the size of subcortical structures. Future investigations could explore subcortical white matter connectivities and hemispheric asymmetries. This approach has previously been conducted on superior longitudinal fasciculus (SLF) (61,62) and holds potential for examining cortico-subcortical connectivity in the context of oscillatory asymmetries.”

(iii) effects not only in basal ganglia and thalamus, but also hippocampus and amygdala (a justification of selection of all ROIs);

We thank the reviewer for this comment. We assessed the hippocampus and amygdala because they are automatically segmented in the FIRST algorithm. As our analysis showed they did not show a relation to the modulation of alpha oscillations, these regions also provide a useful control for our approach. Therefore, we included all subcortical structures in the model and evaluated their predictive impact. This is now addressed in the revised manuscript.

Method (Line 477): “FIRST is an automated model-based tool that runs a two-stage affine transformation to MNI152 space, to achieve a robust pre-alignment of thalamus, caudate nucleus, putamen, globus pallidus, hippocampus, amygdala, and nucleus accumbens based on individual’s T1-weighted MR images.”

Method (Line 576): “The absence of a relationship between modulations of alpha oscillations and the hippocampus and amygdala was expected as these regions typically are not associated with the allocation of spatial attention and thus add validity to our approach. “

(iv) effects that depend on distractor versus target salience (a rationale for the specific two-factor design is missing);

We thank the reviewer for this comment that helped us clarify the manuscript. The two-factor design is to investigate how allocation of attentional resources specifically relates to mechanisms of excitability and suppression mechanism. For this reason, both the salience of the distractor (associated with suppression) and the perceptual load of the target (associated with excitability) had to be manipulated. We clarified the rationale in the revised version as below:

Introduction (Line 96): “We analyzed MEG and structural data from a previous study (27), in which spatial cues guided participants to covertly attend to one stimulus (target) and ignore the other (distractor). To investigate the relationship between the allocation of attentional resources and mechanisms of neural excitability and suppression, the target load and the visual saliency of the distractor were manipulated using a noise mask. This load/salience manipulation resulted in four conditions that affect the attentional demands of target and distractor.”

(v) effects in the absence of reward (why it is important to show that the effect seen previously in a task with reward is seen also in a task without reward);

We thank the reviewer for this clarification comment. We addressed this question in introduction and discussion as below:

Introduction (Line 107): “By examining their role in a task without explicit reward, we aim to elucidate the generalizability of the contributions of subcortical structures to spatial attention modulation. Such a finding would implicate a role for the basal ganglia in cognition beyond the well-studied realm of the estimation of choice values (33). Specifically, in a prior study (28), we observed that the contributions of the basal ganglia were most pronounced when the items in question were associated with a reward. Our current findings broaden our understanding of how subcortical structures are involved in modulating alpha oscillations during top-down spatial attention, in the absence of any reward or value associations. “

Discussion (Line 333): “This convergence of results not only corroborates the validity and consistency of our findings but also extends the empirical foundation supporting the predictive role of the asymmetry of globus pallidus in modulating alpha oscillations beyond reward valence and to the context of attention.”

(vi) effects on rapid frequency tagging.

We thank the reviewer for this constructive comment. We have now included this analysis and added the results to the revised manuscript.

Results (Line 224): “It is worth noting that neither the behavioural nor the rapid invisible frequency tagging (RIFT) measures showed significant relationships with LVs and HLM(α) (Supplementary material, Figure 1 and Table 3).”

Discussion (Line 396): “We did not find any association between the power of RIFT signal and the size asymmetry of subcortical structures. Since to Bayes factors were less than 0.1, we conclude that our RIFT null findings are robust, suggesting a dissociation between how alpha oscillations and neuronal excitability indexed by RIFT relate to subcortical structures.”

Method (Line 548): “We computed the modulation index (MI) for rapid invisible frequency tagging (RIFT) by averaging the power of the signal in sensors on the right when attention was directed to the right compared to when it was directed to the left. This calculation was also performed for sensors on the left. Consequently, we identified the top 5 sensors on each side with the highest MI as the Region of Interest (ROI). Utilizing the sensors within the ROI, we computed hemispheric lateralization modulation (HLM) of RIFT by summing the average MI(RIFT) of the right sensors and the average MI(RIFT) of the left sensors, obtaining one HLM(RIFT) value for each participant. For a more comprehensive analysis, refer to reference (24).”

Supplementary Materials (Line 839): “Figure 1. Lateralization volume of thalamus, caudate nucleus and globus pallidus in relation to hemispheric lateralization modulation of rapid invisible frequency tagging (HLM(RIFT)) on the right and behavioural asymmetry on the left. A and E, The β coefficients for the best model (having three regressors) associated with a generalized linear model (GLM) where lateralization volume (LV) values were defined as explanatory variables for HLM(RIFT) (A) and behavioural asymmetry (E). Error bars indicate standard errors of mean (SEM). B and F, Partial regression plot showing the association between LV_Th_ and HLM(RIFT) (B, p-value = 0.59) and behavioural asymmetry (F, p-value = 0.38) while controlling for LV_GP_ and LV_CN_. C and G, Partial regression plot showing the association between LVGP and HLM(RIFT) (C, p-value = 0.16) and behavioural asymmetry (G, p-value = 0.80) while controlling for LV_Th_ and LV_CN_ . D and H, Partial regression plot showing the association between LV_CN_ and HLM(RIFT) (D, p-value = 0.53) and behavioural asymmetry (H, p-value = 0.74) while controlling for LV_Th_ and LV_GP_. Negative (or positive) LVs indices denote greater left (or right) volume for a given substructure; similarly negative HLM(RIFT) values indicate stronger modulation of RIFT power in the left compared with the right hemisphere, and vice versa; positive behavioural asymmetry value shows higher accuracy when the target was on the right as compared with left, and vice versa for negative behavioural asymmetry values. The dotted curves in B, C, D, F, G, and H indicate 95% confidence bounds for the regression line fitted on the plot in red.

Second, the results are not fully reported. The model space and the results from the model comparison are omitted. Behavioral data and rapid frequency tagging results are not shown. Without having access to the data or the results of the analyses, the reader cannot evaluate whether the null effect corresponds to the absence of evidence or (as claimed in the discussion) evidence of absence.

We thank the reviewer for this constructive suggestion. In the revised manuscript, we incorporated the model space, model comparisons, BIC values from the models, behavioral and rapid frequency tagging analysis methods, and their respective results. Additionally, we computed Bayes factors for our null findings to enhance the interpretability of our results.

Results (Line 199): “This model predicted the HLM(α) values significantly in the GLM (F3,29 = 7.4824, p = 0.0007, adjusted R2 = 0.376) as compared with an intercept-only null model (Figure 4A).”

Although, the β estimate of LV_GP_ only showed a positive trend, removing it from the regression resulted in worse models (AIC and BIC tables in supplementary material).

Supplementary materials (Line 827): “Table 1. Akaike Information Criterion (AIC) and Bayesian Information Criterion (BIC) values for all possible combinations of regressors (Lateralized Volume of subcortical structures). The selected model, with lowest AIC, is marked in green.

**Author response table 1. sa3table1:** 

Regression	Comb. 1	Comb. 2	Comb. 3	Comb. 4	Comb. 5	Comb. 6	Comb. 7	Comb. 8	Comb. 9	Comb. 10	Comb. 11	Comb. 12	Comb. 13	Comb. 14	Comb. 15	Comb. 16	Comb. 17	Comb. 18	Comb. 19	Comb. 20	Comb. 21	Model criterion
	–81.63	–77.78	–73.75	–78.62	–71.94	–73.18	–71.48	NaN	NaN	NaN	NaN	NaN	NaN	NaN	NaN	NaN	NaN	NaN	NaN	NaN	NaN	AIC
–77.33	–73.48	–69.45	–74.32	–67.64	–68.88	–67.18	NaN	NaN	NaN	NaN	NaN	NaN	NaN	NaN	NaN	NaN	NaN	NaN	NaN	NaN	BIC
	–86.72	–79.93	–82.66	–77.88	–78.04	–78.00	–76.42	–81.27	–75.82	–75.83	–73.89	–77.41	–70.41	–71.36	–69.94	–74.79	–76.05	–74.63	–70.08	–68.23	–69.30	AIC
–81.11	–74.32	–77.05	–72.28	–72.44	–72.39	–70.82	–75.66	–70.22	–70.23	–68.29	–71.81	–64.81	–65.75	–64.33	–69.19	–70.44	–69.03	–64.48	–62.63	–63.70	BIC
	–85.30	–87.54	–82.79	–82.96	–82.64	–81.00	–76.18	–76.19	–76.29	–78.86	–79.00	–78.73	–74.25	–74.24	–74.32	–79.67	–74.61	–74.06	–72.55	–78.51	–78.88	AIC
–78.46	–80.71	–75.95	–75.13	–75.81	–74.17	–69.34	–69.35	–69.46	–72.02	–72.17	–71.90	–67.42	–57.40	–67.49	–72.84	–67.77	–67.22	–65.72	–71.68	–72.05	BIC
	–85.52	–81.42	–81.27	–81.21	–83.50	–83.73	–83.30	–79.33	–78.80	–78.85	–77.26	–77.72	–77.09	–72.43	–72.54	–72.49	–75.15	–74.95	–75.04	–70.57	–76.78	AIC
–77.52	–73.43	–73.28	–73.21	–75.50	–75.74	–75.31	–71.33	–70.80	–70.86	–69.27	–69.73	–69.10	–64.44	–64.55	–64.45	–67.15	–66.96	–67.04	–62.58	–68.78	BIC
	–81.51	–81.90	–81.30	–77.65	–77.41	–77.17	–79.95	–79.32	–79.49	–75.29	–73.88	–73.36	–73.75	–68.77	–71.22	–75.31	–72.78	–73.87	–68.85	–72.63	–67.74	AIC
–72.44	–72.83	–72.24	–68.58	–68.34	–68.01	–70.88	–70.25	–70.42	–66.22	–64.81	–64.29	–64.68	–59.70	–62.15	–66.24	–63.71	–64.79	–59.78	–63.56	–58.67	BIC
	–78.13	–77.37	–77.70	–73.63	–75.76	–69.96	–71.25	NaN	NaN	NaN	NaN	NaN	NaN	NaN	NaN	NaN	NaN	NaN	NaN	NaN	NaN	AIC
–68.07	–67.30	–67.63	–63.56	–65.70	–59.89	–61.19	NaN	NaN	NaN	NaN	NaN	NaN	NaN	NaN	NaN	NaN	NaN	NaN	NaN	NaN	BIC
	–74.00	NaN	NaN	NaN	NaN	NaN	NaN	NaN	NaN	NaN	NaN	NaN	NaN	NaN	NaN	NaN	NaN	NaN	NaN	NaN	NaN	AIC
–63.01	NaN	NaN	NaN	NaN	NaN	NaN	NaN	NaN	NaN	NaN	NaN	NaN	NaN	NaN	NaN	NaN	NaN	NaN	NaN	NaN	BIC

**Author response table 2. sa3table2:** 

Regression	Comb. 22	Comb. 23	Comb. 24	Comb. 25	Comb. 26	Comb. 27	Comb. 28	Comb. 29	Comb. 30	Comb. 31	Comb. 32	Comb. 33	Comb. 34	Comb. 35	Modelcriterion
HLM∼,beta_(LV_(1))	NaN	NaN	NaN	NaN	NaN	NaN	NaN	NaN	NaN	NaN	NaN	NaN	NaN	NaN	bar(" AIC ")
	NaN	NaN	NaN	NaN	NaN	NaN	NaN	NaN	NaN	NaN	NaN	NaN	NaN	NaN	BIC
HLM∼,beta_(LV_(1))+,beta_(LV_(2))	NaN	NaN	NaN	NaN	NaN	NaN	NaN	NaN	NaN	NaN	NaN	NaN	NaN	NaN	AIC
	NaN	NaN	NaN	NaN	NaN	NaN	NaN	NaN	NaN	NaN	NaN	NaN	NaN	NaN	BIC
HLMbeta_(LV_(1))+,beta_(LV_(2))+,beta_(LV_(3))	77.3	74.5,9	71.90	71.9,3	73.5,4	75.4	73.45	68.27	66.7	67.5	72.40	70.8,9	72.05	66.3	AIC
	70.5,0	67.7,5	65.0,7	65.0,9	66.7,1	68.5,9	66.6	61.4,3	59.89	60.6,7	65.5,7	64.0,5	65.2	59.4,8	BIC
HLM∼,beta_(LV_(1))+,beta_(LV_(2))+,beta_(LV_(3))+,beta_(LV_(4))	77.76	75.7,6	72.8,3	70.69	70.19	76.7,2	74.4	74.97	70.5,8	71.67	69.66	71.4	64.5,3	68.4	AIC
	69.7,6	67.7,7	64.8,4	62.70	62.1,9	68.7	66.4,9	66.9,8	62.5,9	63.6,8	61.6,7	63.4	56.5,3	60.47	BIC
HLMbeta_(LV_(1))+,beta_(LV_(2))+,beta_(LV_(3))+,beta_(LV_(4))+,beta_(LV_(3))	NaN	NaN	NaN	NaN	NaN	NaN	NaN	NaN	NaN	NaN	NaN	NaN	NaN	NaN	AIC
	NaN	NaN	NaN	NaN	NaN	NaN	NaN	NaN	NaN	NaN	NaN	NaN	NaN	NaN	BIC
HLMbeta_(LV_(1))+,beta_(LV_(2))+,beta_(LV_(3))+,beta_(LV_(4))+,beta_(LV_(5))+,beta_(LV_(n))	NaN	NaN	NaN	NaN	NaN	NaN	NaN	NaN	NaN	NaN	NaN	NaN	NaN	NaN	AIC
	NaN	NaN	NaN	NaN	NaN	NaN	NaN	NaN	NaN	NaN	NaN	NaN	NaN	NaN	BIC
HLMbeta_(LV_(1))+,beta_(LV_(2))+,beta_(LV_(3))+,beta_(LV_(4))+,beta_(LV_(5))+,beta_(LV_(6))+,beta_(LV_(-))	NaN	NaN	NaN	NaN	NaN	NaN	NaN	NaN	NaN	NaN	NaN	NaN	NaN	NaN	AIC
	NaN	NaN	NaN	NaN	NaN	NaN	NaN	NaN	NaN	NaN	NaN	NaN	NaN	NaN	BIC

**Author response table 3. sa3table3:** Bayes factors for correlation between hemispheric laterality of subcortical structures with hemispheric lateralization modulation of rapid invisible frequency tagging (HLM(RIFT)) and with behavioural asymmetry (BA). The Pearson correlation between each subcortical structure with HLM(RIFT) and behavioural asymmetry was calculated. The likelihood of the data under the alternative hypothesis (the evidence of correlation) were subsequently compared to the likelihood under null hypothesis (absence of correlation), given the data. As it is demonstrated in the table, all Bayes factors were below or very close to 1 indicating evidence for the null hypothesis.

Subcortical structure	HLM(RIFT)	BA
Thalamus	0.14	0.20
Caudate Nucleus	0.18	0.15
Putamen	0.14	0.14
Globus Pallidus	0.34	0.13
Hippocampus	0.19	0.25
Amygdala	0.54	1.18
Nucleus Accumbens	0.16	0.17

For the results of frequency tagging signal, we have now included this analysis and added the results to the revised manuscript. We refer the reviewer to our response to the weakness (vi) from reviewer #3.

Third, it remains unclear whether the MMS is the best approach to analyzing effects as a function of target and distractor salience. To address the question of whether the effects of subcortical volumes on alpha lateralization vary with task demands (which we assume is the primary research question of interest, given the factorial design), we would like to evaluate some sort of omnibus interaction effect, e.g., by having target and distractor saliency interact with the subcortical volume factors to predict alpha lateralization. Without such analyses, the results are very hard to interpret. What are the implications of finding the differential effects of the different volumes for the different task conditions without directly assessing the effect of the task manipulation? Moreover, the report would benefit from a further breakdown of the effects into simple effects on unattended and attended alpha, to evaluate whether effects as a function of distractor (vs target) salience are indeed accompanied by effects on unattended (vs attended) alpha.

The reviewer is correct that we did not directly compare between task conditions when we assessed the predictive relationship between basal ganglia lateralization and alpha lateralization. We opted for the multivariate regression approach as this allowed us to simultaneously model the predictive relationship between our continuous predictors and HLM(α) in each condition, allowing us to be most efficient with our level of statistical power (N=33). Indeed, directly comparing between task conditions within one model would result in an extra 16 regressors (1 (intercept) + 4-1 to model the difference between conditions + 3 to model the regressors + 3 x 3 to model each region x task condition interaction). This approach would be underpowered given our sample size, and the ensuing results are likely to be unreliable.

However, we statistically analysed our regression results. Multivariate regression analysis allows one to test the null hypothesis that a given predictor does not contribute to all the dependent variables. A rejection of this hypothesis would suggest that lateralization of a given region of interest significantly predicts variability across all 4 of the task conditions, whereas failure to reject the null would imply that the predictive relationship holds only for that single condition. We tested this global null hypothesis using a MANOVA test and reported the findings in response to weakness two from reviewer #1.

Discussion (Line 384): “In future studies it would be interesting to design an experiment directly addressing which subcortical regions contribute to distractor and target load in terms of modulating the alpha band activity. In order to ensure sufficient statistical power for doing so possibly each factor needs to be addressed in different experiments. “

The fourth concern is that the discussion section is not quite ready to help the reader appreciate the implications of key aspects of the findings. What are the implications for our understanding of the roles of different subcortical structures in the various psychological component processes of spatial attention? Why does the volumetric asymmetry of different subcortical structures have diametrically opposite effects on alpha lateralization? Instead, the discussion section highlights that the different subcortical structures are connected in circuits: "Globus pallidus also has wide projections to the thalamus and can thereby impact the dorsal attentional networks by modulating prefrontal activities." If this is true, then why does the effect of the GP dissociate from that of the thalamus? Also, what is it about the current behavioural paradigm that makes the behavioral readout insensitive to variation in subcortical volume (or alpha lateralization?)?

We thank the reviewer for this feedback. These are indeed all good points, and we hope that our findings will inspire further research to address these issues. In the revised manuscript we now write:

Discussion (Line 349): “The opposite effect of the globus pallidus compared to the thalamus is striking, and possibly explained but the globus pallidus containing GABAergic interneurons. Thus the inhibitory nature of the globus pallidus projections to thalamus could explain why they are related to the alpha modulation in different manners (57).”

Discussion (Line 379): “Moreover, the current study faced methodological constraints, limiting the analysis to the entire thalamus. […] . It would be of great interest to conduct further investigations to quantify the distinct impacts of individual thalamic nuclei on the association between subcortical structures and the modulation of oscillatory activity.“

Discussion (Line 388): “Moreover, our failure to identify a relationship between the lateralized volume of subcortical structures and behavioural measures should be addressed in studies that are better designed to capture performance asymmetries (63). Individual preferences toward one hemifield, which were not addressed in the current study design, could potentially strengthen the power to detect correlations between structural variations in the subcortical structures and behavioural measures.”

**Recommendations for the authors:**

**Reviewer #1 (Recommendations For The Authors):**
Minor comment:Between-subject correlation/regression analyses always rely on the assumption that the underlying dependent measures are reliable. While the reliability of asymmetries of subcortical structures can be assumed, the reliability of lateralized alpha oscillations during spatial attention can be questioned. It would be helpful if the authors could test the reliability of alpha lateralization, for instance by calculating HLM(a) in the first and second half of the experiment and correlating the resulting HLM(a) values (split-half reliability).

We appreciate the reviewer for their insightful comment. Acknowledging that the between-subject regression relies on the reliability of alpha lateralization. Nonetheless, a previous study has demonstrated consistent results regarding HLM(α). We have further elaborated on these aspects in the discussion section:

Discussion (Line 328): “Furthermore, our regression analysis outcomes align with the findings of Mazzetti et al. (28) underscoring the significant predictive influence exerted by the lateralized volume of globus pallidus on the modulation of hemispheric lateralization in alpha oscillations during spatial attention tasks. This convergence of results not only corroborates the validity and consistency of our findings but also extends the empirical foundation supporting the predictive role of the asymmetry of globus pallidus in modulating alpha oscillations within the context of attention.”

**Reviewer #3 (Recommendations For The Authors):**
We recommend that a revised version of the manuscriptClarifies the theoretical basis for the 6 key design & analysis choices that we have outlined above;

We thank the reviewer for their precision. We addressed the concerns outlined above in the previous section.

Also clarifies the task description (perhaps referring to target and distractor salience instead of target load versus distractor salience might help);

Thank you for this constructive comment. We used the terms ‘load’ for target and ‘salience’ for distractor because the noise manipulation of the faces reduces the salience of the image which results in distractors being less distractive (easier) but targets being more perceptually loaded (harder). The explanation of these terms is made clear in the revised manuscript.

Method (Line 447): “Over trials, the perceptual load of targets was manipulated using a noise mask; noisy targets are harder to detect than clear targets and therefore incur greater perceptual load in their detection. The saliency of distractor stimuli was also manipulated using a noise mask; noisy distractor stimuli are less salient than clear distractors and therefore less disruptive to performance on the detection task. The noise mask was created by randomly swapping 50% of the stimulus pixels (Figure 1B). This manipulation resulted in four target-load/distractor-saliency conditions: (1) target: low load, distractor: low saliency (i.e., clear target, noisy distractor), (2) target: high load, distractor: low saliency (i.e., noisy target, noisy distractor), (3) target: low load, distractor: high saliency (i.e., clear target, clear distractor), (4) target: high load, distractor: high saliency (i.e., noisy target, clear distractor) (Figure 1B and C).”

Fully reports all the data, including those of the model comparisons, the behavioural results, and the rapid frequency tagging results;

We thank the reviewer for this constructive comment. We refer the reviewer to our response to second comment and comment (vi) from reviewer #3.

Reports interaction effects to directly test the modulating role of task demands in the link between volume and alpha, and break down the alpha lateralization indices into their simple effects on the ipsilateral and contralateral hemispheres;

task demands have been addressed in response to in response to weakness two from reviewer #1.

Regarding the second part of the comment, in our study, to compare the lateralized modulation of alpha oscillations between the right and left hemispheres, we computed hemispheric lateralization modulation. This involved dividing trials into attention right and attention left. Subsequently, we calculated the lateralization index separately for sensors on the right and left. Specifically, this entailed computing ipsilateral – contralateral for sensors on the right and contralateral – ipsilateral for sensors on the left side of the brain. We addressed this concern in methods section as below:

Method (Line 537): “As MI(α) consistently represents power of alpha in attention right versus attention left conditions, it entails the comparison between ipsilateral and contralateral alpha modulation power for sensors located on the right side of the head. The same comparison applies inversely for sensors situated on the left side of the brain.”

Clarifies in the discussion section the specific implications of the results for our understanding of the link between distinct subcortical structures and distinct component processes of spatial attention.

We thank the reviewer for their constructive comment. This point is addressed in response to the fourth concern of reviewer #3.

More detailed specific recommendations are provided below:Line 40ff: In this paragraph, the theoretical framework concerning the function of the subcortical regions of interest is described. Here, the authors jump back and forth between the role of the basal ganglia and the role of the thalamus. For clarity, we would advise to describe the functions of these two structures one after the other. And include a justification for assessing the hippocampus and the amygdala.

We appreciate the reviewer’s preciseness in this comment. We put the description of these structures one after the other in the revised manuscript as below:

Introduction (Line 44): “For instance, it has been shown that the pulvinar plays an important role in the modulation of neocortical alpha oscillations associated with the allocation of attention (9). Studies in rats and non-human primates have shown that both the thalamus and superior colliculus, are involved in the control of spatial attention by contributing to the regulation of neocortical activity (9-11). Notably, when the largest nucleus of the thalamus, the pulvinar, was inactivated after muscimol infusion, the monkey’s ability to detect colour changes in attended stimuli was lowered. This behavioral deficit occurred when the target was in the receptive field of V4 neurons that were connected to lesioned pulvinar (12). The basal ganglia play a role in different aspects of cognitive control, encompassing attention (13,14), behavioural output (15), and conscious perception (16). Moreover, the basal ganglia contribute to visuospatial attention by linking with cortical regions like the prefrontal cortex via the thalamus.”

Justification for assessing the hippocampus and the amygdala has been addressed in response to weakness (iii) from reviewer #3.

The authors mention they defined symmetric clusters of 5 sensors in each hemisphere that showed the highest modulation, but it is not clear how this number of sensors was determined a priori.

We thank the reviewer for their comment. We edited the revised manuscript as below:

Method (Line 536): “Ten sensors were selected to ensure sufficient coverage of the region exhibiting alpha modulation as judged from prior work (62).”

In line 141, the abbreviation HLM is first mentioned but the concept of "hemispheric lateralization modulation of alpha power" is only mentioned in the following section. For the ease of the reader, the abbreviation could be mentioned together with this concept at the beginning of this paragraph.

We thank the reviewer for the attention. In the revised manuscript HLM(α) is now mentioned with its concept.

Results (Line 153): “Next, we computed the hemispheric lateralization modulation of alpha power HLM(α) in each individual.”

In line 188 of the results section, it is mentioned that the table including the AIC values for model comparisons is in the supplementary material, however, we could not locate this table.

We thank the reviewer for their constructive feedback. The supplementary materials were uploaded in a separate file, and it must not have been available to the reviewers. We have now added the supplementary materials to the end of the manuscript for convenience.

Figure 4 is missing the panel headers A, B, C, and D.

We thank the reviewer for their precision. This figure is now fixed.

**Author response image 2. sa3fig2:** 

In lines 205 and 206, behavioral and rapid frequency tagging analysis are mentioned. For the behavioral analysis, the method is described, but no results are provided. For the rapid frequency tagging, neither the methods nor the results are described. To evaluate the strength of this (non)-evidence, we would advise to elaborate on these analysis steps and report the results in the supplementary material.

We thank the reviewer for this constructive comment. A brief explanation of the analysis method of rapid frequency tagging signal is added to the revised manuscript.

Method (Line 548): “We computed the modulation index (MI) for rapid invisible frequency tagging (RIFT) by averaging the power of the signal in sensors on the right when attention was directed to the right compared to when it was directed to the left. This calculation was also performed for sensors on the left. Consequently, we identified the top 5 sensors on each side with the highest MI as the Region of Interest (ROI). Utilizing the sensors within the ROI, we computed hemispheric lateralization modulation (HLM) of RIFT by summing the average MI(RIFT) of the right sensors and the average MI(RIFT) of the left sensors, obtaining one HLM(RIFT) value for each participant. For a more comprehensive analysis, refer to reference (24).”For a more detailed answer, we refer the reviewer to the second comment from reviewer #3.

For the paragraph starting at line 209, we would recommend referring to Figure 1.

We thank the reviewer for their suggestion. This paragraph is now referring to Figure 1.

Results (Line 229): “To relate load and salience conditions of the task to the relationship between subcortical structures and the alpha activity, we combined low-load or high-load targets with high-saliency or low-saliency distractors to manipulate the perceptual load appointed to each trial (Method section, Figure 1). “

Figure 5 as well as the report of the β weights in this section shows a difference in the direction of the effect for the thalamus compared to the globus pallidus and caudate nucleus which is not discussed in this section.

We thank the reviewer for bringing this important point to our attention. We addressed this comment in the discussion section as below:

Discussion (Line 349): “The opposite effect of the globus pallidus compared to the thalamus is striking, and possibly explained by the globus pallidus containing GABAergic interneurons. Thus the inhibitory nature of the globus pallidus projections to thalamus could explain why they are related to the alpha modulation in different manners (54).”

Discussion (Line 379): “Moreover, the current study faced methodological constraints, limiting the analysis to the entire thalamus. […] It would be of great interest to conduct further investigations to quantify the distinct impacts of individual thalamic nuclei on the association between subcortical structures and the modulation of oscillatory activity.“

Comment 2 on line 80 is addressed in the paragraph following 264 by describing volumetric changes in basal ganglia in neurodegenerative disorders such as PD or Huntington's. Still, the link of how a decrease in volume in this region could be causally linked to changes in alpha-band power could be better supported.

We thank the reviewer for their constructive feedback. We are here highlighting the significant correlation between subcortical structures and changes in attention modulated alpha oscillation. We added a few more references to the discussion supporting the relationship between size and function in relation to neurological disorders. We also edited the manuscript to make this point clearer as below:

Introduction (Line 113): “Our current findings broaden our understanding of how subcortical structures are involved in modulating alpha oscillations during top-down spatial attention, independent of any reward or value associations. “

Discussion (Line 305): “Changes in neocortical oscillatory activity have also been observed in neurological disorders which mainly are known to affect subcortical structures. For example, individuals with Alzheimer's Disease demonstrate an increase in slow oscillatory activities and a decrease in higher frequency oscillations (42). Moreover, in patients with Parkinson’s Disease, the power of beta oscillations increases relatively to when they are dopamine-depleted compared with when they are on dopaminergic medication (43). “

Related to the previous comment on behavioral and rapid frequency tagging results, these are difficult to evaluate without mention of the methods and/or results.

We thank the reviewer for this comment. We refer the reviewer to our response to the second comment from reviewer #3.

The authors show differential effects of target load and distractor saliency; however, we missed the description of how these two variables differ conceptually as they are both described as contributing to task difficulty and it is not described why we would expect differential effects for these concepts (or in other words, how the authors explain the differential effects).

We thank the reviewer for their comment. Directly comparing between task conditions within one model would result in an extra 16 regressors (1 (intercept) + 4-1 to model the difference between conditions + 3 to model the regressors + 3 x 3 to model each region x task condition interaction). Give our sample size, this study is underpowered to directly compare alpha lateralisation in contralateral versus ipsilateral conditions. For a more detailed answer please refer to our response to weakness two from reviewer #1.

Line 364ff: Based on the description of the experimental design, it is not clear to us whether participants only had to report on the change in gaze for the stimulus in the cued hemifield.

We thank the reviewer for this comment, which prompted us to clarify the experimental design as below:

Method (Line 440): “Then followed a 1000 ms response interval where participants were asked to respond with their right or left index finger whether the gaze direction of the cued face shifted left or right.”

Line 47ff: As mentioned above, the AIC table is not included. Further, as it is mentioned that BIC values led to similar results (indicating that they are not identical), it would be valuable to report both AIC and BIC values.

We thank the reviewer for their constructive feedback. The supplementary materials were uploaded in a separate file, and it must not have been available to the reviewers. We have now added the BIC values and attached the supplementary materials to the end of the manuscript for convenience.